# Correct regionalization of a tissue primordium is essential for coordinated morphogenesis

Yara E Sánchez-Corrales[1,2]*, Guy B Blanchard[3], Katja Röper[1]*

[1]MRC Laboratory of Molecular Biology, Cambridge Biomedical Campus, Cambridge, United Kingdom; [2]Genetics and Genomic Medicine Programme, Institute of Child Health, University College London, London, United Kingdom; [3]Department of Physiology, Development and Neuroscience, University of Cambridge, Cambridge, United Kingdom

**Abstract** During organ development, tubular organs often form from flat epithelial primordia. In the placodes of the forming tubes of the salivary glands in the *Drosophila* embryo, we previously identified spatially defined cell behaviors of cell wedging, tilting, and cell intercalation that are key to the initial stages of tube formation. Here, we address what the requirements are that ensure the continuous formation of a narrow symmetrical tube from an initially asymmetrical primordium whilst overall tissue geometry is constantly changing. We are using live-imaging and quantitative methods to compare wild-type placodes and mutants that either show disrupted cell behaviors or an initial symmetrical placode organization, with both resulting in severe impairment of the invagination. We find that early transcriptional patterning of key morphogenetic transcription factors drives the selective activation of downstream morphogenetic modules, such as GPCR signaling that activates apical-medial actomyosin activity to drive cell wedging at the future asymmetrically placed invagination point. Over time, transcription of key factors expands across the rest of the placode and cells switch their behavior from predominantly intercalating to predominantly apically constricting as their position approaches the invagination pit. Misplacement or enlargement of the initial invagination pit leads to early problems in cell behaviors that eventually result in a defective organ shape. Our work illustrates that the dynamic patterning of the expression of transcription factors and downstream morphogenetic effectors ensures positionally fixed areas of cell behavior with regards to the invagination point. This patterning in combination with the asymmetric geometrical setup ensures functional organ formation.

## Editor's evaluation

This paper addresses a fundamental question in developmental biology, that is, how morphogenetic movements driving tissue folding are patterned to occur with the correct spatiotemporal dynamics. By correlating dynamic patterns of transcription factor expression with rigorous, quantitative analyses of cell behaviors across the salivary gland primordium in *Drosophila*, their results suggest Hkb and Fkh transcription factor patterning induces switches in cell behaviors at fixed positions to promote continued morphogenesis of the tubular structure. This mechanism is likely to be more generally important for the development of complex tubular organs.

*For correspondence:
y.sanchez-corrales@ucl.ac.uk
(YES-C);
kroeper@mrc-lmb.cam.ac.uk (KR)

**Competing interest:** The authors declare that no competing interests exist.

## Introduction

Complex three-dimensional organs arise from simple tissue primordia, and in many cases, these primordia are flat polarised epithelial sheets. Early in development the expression of the first patterning genes broadly sets up embryonic regions. This is followed by the activation of gene regulatory networks that specify the fate of tissue primordia in defined locations (*Gilmour et al., 2017*; *Sidor and Röper, 2016*). The patterning and fate determination gene products do not directly affect morphogenetic changes, but rather instruct the expression of downstream morphogenetic effectors that drive a tissue primordium down a path of defined physical changes. In many well-studied cases of tissue morphogenesis, such as mesoderm invagination or germband extension in the fly, many morphogenetic effectors are induced evenly across the tissue primordium. Regional differences in the physical changes or 'behaviors' of cells observed in these primordia can arise due to physical effects and feedback or interference either from cells within the primordium or from surrounding tissues (*Chanet et al., 2017*; *Collinet et al., 2015*; *Lye et al., 2015*). A further layer of control and complexity is added as the primordia themselves can also be patterned to guide the differential behavior and changes of groups of cells within a primordium. Such pre-patterning of morphogenetic events within a single tissue primordium is much less understood.

We use the formation of a narrow-lumen tube in the *Drosophila* embryo as a model system to identify key requirements for successful organ formation. The symmetrical tubes of the embryonic salivary glands form from a flat, nearly circular epithelial primordium, the salivary gland placode (*Figure 1A*). We recently uncovered a patterning of key cell behaviors that drive the initiation and earliest stages of the tube budding process (*Gillard et al., 2021*; *Sanchez-Corrales et al., 2018*). Interestingly, the point of invagination from the circular placode primordium is not in the center of the placode, but in an asymmetric, eccentric position in the dorsal-posterior corner (*Figure 1A*). Using quantitative morphometric methods, we showed that the early apical constriction and associated cell wedging at the position of the invagination pit initiates the tissue bending (*Booth et al., 2014*; *Sanchez-Corrales et al., 2018*). Isotropic constriction driven by apical-medial actomyosin is concentrated near the forming invagination pit (*Figure 1A' and A''*), but at a distance to the pit a second cell behavior dominates cell intercalation. Driven by a polarised junctional accumulation of actomyosin, cell intercalation events such as T1 exchanges and rosette formation and resolution help to elongate the tissue radially toward the invagination pit and contract it circumferentially, via circumferential neighbor gains (*Figure 1A'*; *Sanchez-Corrales et al., 2018*).

This asymmetric setup of the placode raises two questions: first, is the asymmetry required for successful organ formation or could a symmetrical setup not lead to the same result? An indication that the asymmetry of the salivary gland placode is important for wild-type tube formation is given by the reports that in certain mutant situations where a more central invagination appears to form, the overall morphogenesis is disrupted and the invaginated structure that forms is sack-like and irregular. This is true for embryos lacking the transcription factor Hkb that is expressed early in the placode (*Myat and Andrew, 2000b*; *Myat and Andrew, 2002*), and could therefore potentially implicate Hkb in establishing early placode patterning.

Second, how is this asymmetry of the placode established prior to morphogenesis commencing and then maintained throughout the process? Previous studies have shown that several proteins known to be expressed in the placode can be detected at the mRNA level initially in the dorsal posterior region of the placode. These include the transcription factor Forkhead (Fkh) (*Myat and Andrew, 2000a*), the dynein-associated protein Klar (*Myat and Andrew, 2002*), as well as the kinase Btk29 (*Chandrasekaran and Beckendorf, 2005*). Defined expression of these and other factors at the dorsal-posterior corner could set up the initiation of cell shape changes in this region. In agreement with this, we previously found that $fkh^{-/-}$ mutants specifically fail to apically constrict in the dorsal-posterior corner, though they do show intercalation behavior, and these mutants do not invaginate a tube (*Sanchez-Corrales et al., 2018*). Following on from the early regionalized behaviors that drive the initial tissue bending and tube invagination (*Sanchez-Corrales et al., 2018*), it is thus far unclear what mechanisms continue to drive the sustained invagination until all secretory cells have internalized. This could be due to continued transcriptional patterning and activation of downstream effectors, and/or could be due to mechanical signaling and feedback.

Here, we investigate how the asymmetry is set up and maintained during salivary gland tube formation, in particular beyond the initial stages. We found that apical constriction is always strongest

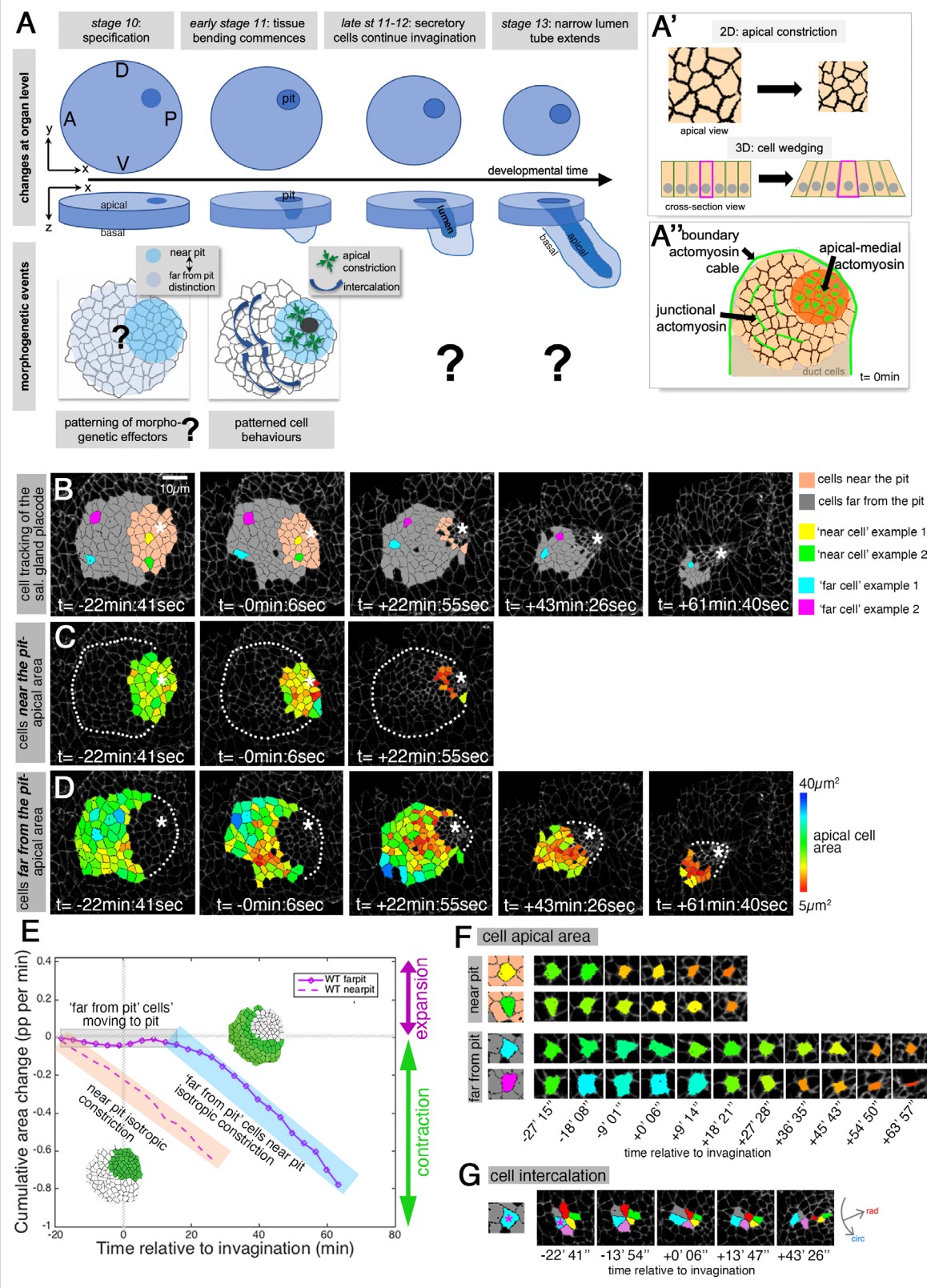

**Figure 1.** Patterned apical constriction remains fixed around the pit over time. (**A**) The tubes of the salivary glands form from a flat epithelial placode with a circular geometry. Cells invaginate through an asymmetrically positioned invagination pit at the dorsal-posterior corner, embryonic axes of anterior-posterior (AP) and dorso-ventral (DV) are indicated. In the placode at early stages cell behaviors are highly patterned, with cells near the pit predominantly constricting isotropically and cells far from the pit predominantly intercalating (**Sanchez-Corrales et al., 2018**). (**A'**) Apical constriction

*Figure 1 continued on next page*

*Figure 1 continued*

within the apical domain of an epithelial cell, a 2D change, equates to a behavior of cell wedging within the 3D context of a whole cell, as shown in a cross-sectional view. (**A″**) Patterned cell behaviors are driven by distinct pools of apical actomyosin. Three pools of actomyosin can be distinguished in the placode: apical-medial actomyosin in cells near the invagination pit (dark orange), leading to isotropic apical constriction; polarised junctional actomyosin, driving the initiation of directed cell intercalation events in cells further away from the pit (light orange); a circumferential actomyosin cable at the boundary of the placode. (**B**) Stills of a representative segmented time-lapse movie, color-coded to indicate cell near (pink) and far (gray) from the invagination pit, with individual cell examples in both regions highlighted. As calculated previously (*Sanchez-Corrales et al., 2018*), we employ a radial coordinate system centered at the invagination point at t=0 min. We specify the 'near to the pit' region as the region located between the invagination point and up to 33% of the distance between the invagination point and the far edge of the placode (with this region usually being within 13.7±3 μm of the pit), and with the rest of the cells defined as 'far from the pit.' The cell outlines in the time-lapse movie are labeled using Armadillo-YFP. These and all confocal panels shown in the figures are always oriented with anterior to the left and dorsal up. See *Figure 1—video 1*. (**C, D**) Placodal cells visually split into near the pit (**C**) and far from the pit (**D**) cells and color-coded for apical area size. The asterisks in (**B–D**) indicate the (future) position of the invagination pit, white dotted lines indicate the boundary of the placode in regions where cells are not highlighted. (**E**) Analysis of apical area change expressed in proportion (pp) per minute, pooled from seven time-lapse movies for cells that were specified at t=0 min to be located near the invagination pit (dashed line) and cells that were specified at t=0 min to be located far from the invagination pit (solid line). Wild-type movies were aligned in time using as t=0 min the frame just before the first sign of invagination at the future tube pit was evident. Note that cells near the pit continuously constrict until internalized (pink shaded box), whereas cells far from the pit do not constrict significantly until t=20 min whilst moving closer to the pit (gray shaded box), but once they are in close proximity to the pit they also constrict until internalized (blue shaded box). Details of cell numbers and lengths of movies analyzed are shown in *Figure 1—figure supplement 1*. (**F**) Apical area size evolution of individual cell examples of cells near the pit and far from the pit as highlighted in color in (**B**). (**G**) Example of a cell from the far from the pit region (blue cell in (**B**) and (**F**)) undergoing a cell intercalation process as part of a rosette of cells that contracts circumferentially and resolves to elongate the tissue radially toward the invagination point.

The online version of this article includes the following video and figure supplement(s) for figure 1:

**Figure supplement 1.** Patterned apical constriction remains fixed around the pit over time.

**Figure 1—video 1.** Contiunous apical constriction near the pit and switch to apica constriction of cell approaching the pit.

https://elifesciences.org/articles/72369/figures#fig1video1

near the invagination pit at any moment in time. As more and more coronae of cells approach the region near the pit, they switch behavior from predominantly intercalating to predominantly apically constricting/wedging. This continued switch to apical constriction leads to a smooth continued invagination process akin to a 'standing wave' of apical constriction, through which the placodal cells flow. The continued near-pit constriction is driven by apical-medial myosin, whose peak intensity and peak pulsatile strength is tracked in a fixed position close to the pit over time. Using live-imaging of endogenous fluorescently tagged versions of Hkb and Fkh, we show that their expression levels pre-pattern where initial constriction is occurring. Both are upstream of a dynamic patterning of the GPCR ligand Fog, that in turn activates apical-medial myosin. Loss of the correct pre-patterning, such as in $hkb^{-/-}$ mutants or when Fog is overexpressed, leads to loss of the symmetrical final organ shape with a narrow lumen and instead gives rise to expanded and sack-like glands with widened lumena.

Our work establishes that after the tissue is patterned at the posterior corner, cells follow dynamic rules of behavior as they approach the pit. Symmetrical tube morphogenesis thus relies on a regionalization and patterning of the primordium long before cells show distinctive behaviors. Establishing general mechanisms that modulate the orderly and repetitive behavior of cells to form correctly shaped simple model organs such as the salivary glands will assist the understanding of the morphogenesis of other more complex tubular organs in mammals.

## Results

### Patterned apical constriction remains fixed around the pit over time

The initial morphogenetic change that occurs during the budding of the tube of the salivary glands in the *Drosophila* embryos is the apical constriction of cells at the point of the future site of invagination, the pit (*Figure 1A and A'*; *Girdler and Röper, 2014*; *Myat and Andrew, 2000a*; *Myat and Andrew, 2000b*; *Sanchez-Corrales et al., 2018*). We recently provided the first quantitative morphometric analysis of the very early stages of this process, and revealed that, first, isotropic apical constriction near the future pit commences long before changes at the tissue level are apparent, and that, second, the apical constriction is indicative of a 3D cell behavior of wedging of cells at the pit (*Figure 1A'*). Importantly, we uncovered a strong regionalization of cell behaviors during the early stages of tube

budding: cells near the invagination point predominantly showed apical constriction behavior while cells at a distance from the pit predominantly showed intercalation behavior (*Sanchez-Corrales et al., 2018*; *Figure 1A*). What is unclear is how the early tube initiation via isotropic apical constriction of cells, combined with directional intercalation of the cells still positioned away from the forming pit, evolves over time. The process of tube formation is characterized by the continuous invagination of cells and the formation of a symmetrical narrow-lumen tube from an asymmetric placode primordium. We can imagine two scenarios: in one, cell behavior is fixed at early stages of budding, that is, apical cell constriction is restricted to the pit and only apparent during early stages. In a second scenario, cell behavior depends on a cell's position relative to the invagination pit and thus cell behaviors change as cells move toward the invagination point.

To distinguish between these scenarios, we segmented and tracked placodal cells from time-lapse movies covering a –40 min to +70 min time interval of the invagination process, with t=0 min defined as the first occurrence of tissue bending (see *Figure 1—figure supplement 1* for time intervals of analyzed movies and cell numbers analyzed) and performed quantitative analyses (*Figure 1B and C*). At the latest time point covered, the majority of secretory cells had completely internalized and formed the internal tube. As calculated previously (*Sanchez-Corrales et al., 2018*), we employ a radial coordinate system centered at the invagination point at t=0 min. We specify the 'near to the pit' region as the region located between the invagination point and up to 33% of the distance between the invagination point and the far edge of the placode (with this region usually being within 13.7±3 μm of the pit), and with the rest of the cells defined as 'far from the pit.'

Cells positioned near the pit at t=0 min (*Figure 1B*, pink) constricted their apices isotropically and rapidly and were internalized by about t=+ 20 min (*Figure 1B and C* and dashed curve in *Figure 1E*). Cells positioned far from the pit at t=0 min (*Figure 1B*, gray) and moving into a position closer to the invagination point began constricting (*Figure 1B and D* and solid curve in *Figure 1E*; *Figure 1—video 1*), suggesting that cells changed their behavior depending on where they were positioned in the placode with respect to the invagination point. Cells near the pit displayed isotropic constriction and thus a negative cumulative apical area change from the start (*Figure 1E*, pink highlighted), whereas cells initially positioned far from the pit did not change their apical area over the first ~20 min (*Figure 1C*, gray highlighted), but then began to display isotropic apical constriction (*Figure 1E*, blue highlighted). The cell behaviors and changes of cell behaviors were very apparent when we analyzed individual cells: cells located in the near pit region (*Figure 1F*, yellow and green cells) constricted progressively, whereas cells located in the region far from the pit at t=0 min (*Figure 1F*, blue and magenta cells) retained their apical area initially and only began constricting from t>+20 min onwards. As described previously (*Sanchez-Corrales et al., 2018*), cells far from the pit initially showed predominantly a cell intercalation behavior that constricts the tissue primordium circumferentially whilst expanding it radially, thereby moving cells closer to the invagination pit (*Figure 1G*). As they approached the invagination pit, these cells also started to apically constrict (*Figure 1E* solid line, highlighted in blue and *Figure 1F*).

These data strongly suggest that an initially pre-patterned behavior, isotropic constriction at the pit and intercalation further away, is dynamically adjusted during tube budding from a flat epithelial sheet, so that cells within a similar distance to the invagination pit display the same behavior.

## Sustained apical constriction near the pit is driven by sustained apical-medial myosin

Early apical constriction at the pit is driven by a highly dynamic apical-medial actomyosin network (*Booth et al., 2014*; *Sanchez-Corrales et al., 2018*). We next sought to determine whether the apical-medial actomyosin dynamics would differ among regions in the placode. We analyzed apical-medial actomyosin levels in cells across the placode, and firstly illustrate our findings in two snapshots: cells near the future invagination point at t=0 min showed strong accumulation of apical-medial myosin (*Figure 2A, A″ and C*; see also *Figure 2—video 1*), whereas cells far from the pit and close to the anterior placode boundary (*Figure 2A and A'*) showed mainly junctional myosin II accumulation. At t=30 min, the cells now located right next to the invagination pit continued to display strong apical-medial myosin accumulation (*Figure 2B and B″*), but in addition cells that were initially 'far from the pit,' but were now in a closer proximity to the pit though still near the anterior boundary, now also started to display apical-medial actomyosin (*Figure 2B and B'*). Thus, being moved into a position

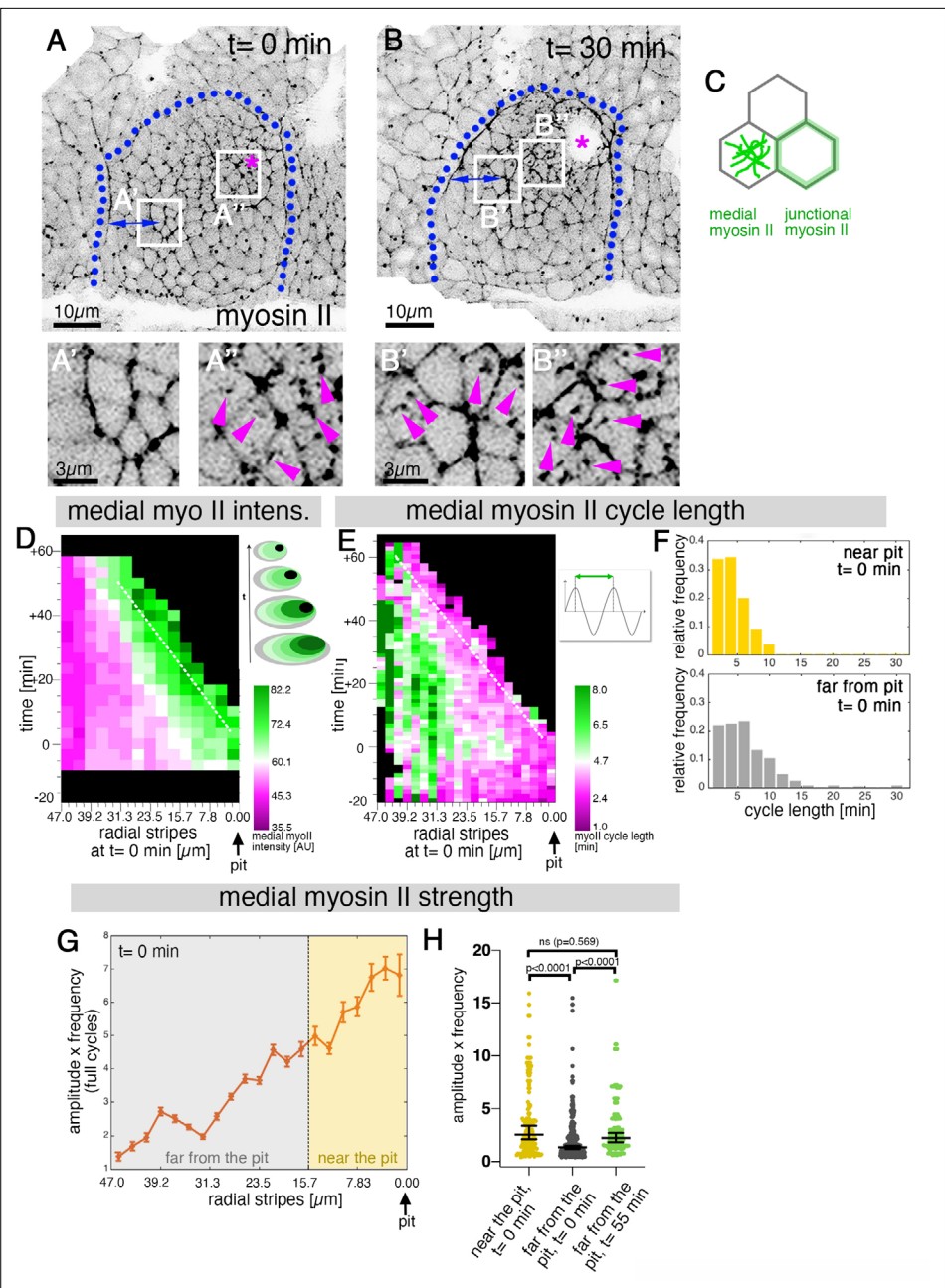

**Figure 2.** Sustained apical constriction near the pit is driven by apical-medial myosin. (**A-B″**) Still images of a time-lapse movie of embryos expressing SqhGFP to label myosin II at t = 0 min (**A-A″**) and t = 30 min (**B-B″**). White boxes indicate the corresponding higher magnifications shown below, with (**A″,B″**) showing cells that are at these timepoints located near the pit, and (**A′,B′**) showing cells near the boundary (small blue double arrows indicate similar closeness to the boundary). Magenta asterisk marks the invagination pit, blue dotted lines mark the boundary of the placode. (**C**) Schematic apical view of epithelial cells, illustrating junctional and apical-medial pools of actomyosin. Arrows in (**A″, B′, B″**) point to apical-medial actomyosin. See also *Figure 2—video 1*. (**D**) Spatial representation of the average medial myosin intensity from an exemplary time lapse movie, with radial location (collapsed into stripes) of cells specified at t = 0min. The dashed white line marks the peak of medial myosin intensity that is always adjacent to the invagination pit as cells flow into it over time. The schematic illustrates the evolution of the radial stripes analysed, with stripes initially close to the pit internalised first and hence a position close to the pit 'moving' across this type of plot diagonally in wild-type placodes. (**E**) Spatial representation of the distribution of the myosin cycle length across the salivary gland placode, with radial location (collapsed into stripes) of cells specified at t = 0 min. The dashed line marks the region of shortest cycle length

*Figure 2 continued on next page*

*Figure 2 continued*

that is always adjacent to the invagination pit as cells flow into it over time. The mean of 3 movies is shown. The schematic illustrates one cycle of a myosin pulsation defined as periodic increases and decreases in medial myosin II intensity. (**F**) The distribution of myosin II cycle lengths between cells near the pit and far from the pit varies: the median of the cycle length for cells near the pit is 3.7 min (n = 146) and SD+/- 2.08 min, while cells far from the pit have a median myosin II cycle length of 4.64 min (n = 301) and SD+/- 2.72 min. Thus, the cells that are located far from the pit show longer cycle length (p < 0.0001 Mann Whitney test). Data is pooled from 3 movies at t = 0 min. (**G**) Spatial representation of the strength of myosin oscillations in radial stripes from the pit (0 µm) to the boundary of the placode (47µm) at t = 0 min. Mean and standard error of the mean are shown. Data are pooled from 3 movies ranging from -7.5-56.25 min; -16.76 to 55.48 min and -17.42 to 67.26 min. Regions corresponding to cells near the pit and far from the pit are indicated by coloured shading. (**H**) Medial myosin strength, expressed as the product of amplitude x frequency of the oscillations, for all cells analysed from 3 movies, split into cells located near to the pit (n=148) compared to cell located far from the pit (n = 302) at t = 0 min from 3 movies. Cells near the pit have significantly higher medial myosin strength than cells located far from the pit; statistical significance as determined (Mann Whitney test, with p<0.0001). Cells far from the pit at later time points (t = 55-60 min, n = 134) increased the medial myosin strength and were not significantly different from cells near the pit at t = 0 min (Mann Whitney test, p = 0.5690). In all analyses, t = 0 min is defined as the frame just before the first sign of invagination at the future pit was evident. See also *Figure 2—figure supplement 1* and *Figure 2—video 1*.

The online version of this article includes the following video and figure supplement(s) for figure 2:

**Figure supplement 1.** Sustained apical constriction near the pit is driven by apical-medial myosin.

**Figure 2—video 1.** Continuous medial myosin oscillations near the pit.

https://elifesciences.org/articles/72369/figures#fig2video1

---

near the invagination pit appeared to determine levels of apical-medial myosin accumulation. Looking placode-wide over the whole time period analyzed, the peak of the apical-medial myosin fluorescence intensity tracked in a near-fixed position to the edge of the pit over time (*Figure 2D*, white dotted line). The location of the highest medial myosin intensity was also where cells showed the shortest cycle lengths of medial myosin II fluctuations (*Figure 2E*, white dotted line). Interestingly, cells near the pit at t=0 min showed an overall shorter myosin II cycle length (the time period elapsed between two peaks of highest myosin II intensity during the oscillation, see schematic in *Figure 2E*) compared to cells located far from the pit (*Figure 2F*). As longer cycle length correlates with unproductive myosin cycles (*Booth et al., 2014*), this was in line with cells further away from the pit showing less apical constriction. Furthermore, the strength of myosin oscillations can be expressed as the product of the amplitude and frequency of the oscillation (*Booth et al., 2014*). This myosin strength across the placode at t=0 min was much higher in cells near the invagination point when plotted against the distance to the pit, with a gradual decrease in myosin strength toward the placode boundary (*Figure 2G*). However, cells originally far from the pit at t=0 min increased their myosin strength over time (and concomitantly decreased their cycle length; see *Figure 2—figure supplement 1*) whilst moving into a position closer to the pit, so that by t=55 min their myosin strength had increased to levels previously shown by cells near the pit at t=0 min (*Figure 2H*).

Thus, our data show that the early asymmetric setup of high apical-medial myosin II intensity and activity that was clustered near the pit was maintained during the continued invagination of the tube, in turn driving the continued apical constriction near the pit.

## *hkb*[-/-] mutants show a delayed symmetrical apical constriction

In order to uncover the role of the initial asymmetric patterning of cell behaviors within the placode, we turned our attention to mutant situations: we already knew that the transcription factor Fkh was important for invagination, as *fkh*[−/−] mutants do not show localized apical constriction or any invagination (*Myat and Andrew, 2000a*; *Sanchez-Corrales et al., 2018*). A possibly more interesting situation is present in a previously published mutant of the transcription factor Huckebein (Hkb) that was reported to show a central invagination pit, combined with malformed invaginated salivary glands at later stages (*Myat and Andrew, 2000b*; *Myat and Andrew, 2002*).

We collected and segmented time-lapse movies of *hkb*[−/−] mutant embryos over the same time period as the control wild-type embryos (*Figure 3* and *Figure 3—figure supplement 1*; *Figure 3—videos 1–3*). Whereas in wild-type embryos, at t ~ +40 min, most secretory cells of the salivary gland placode have already invaginated (*Figure 3A‴, B‴ and C‴*), in the *hkb*[−/−] mutants no invagination

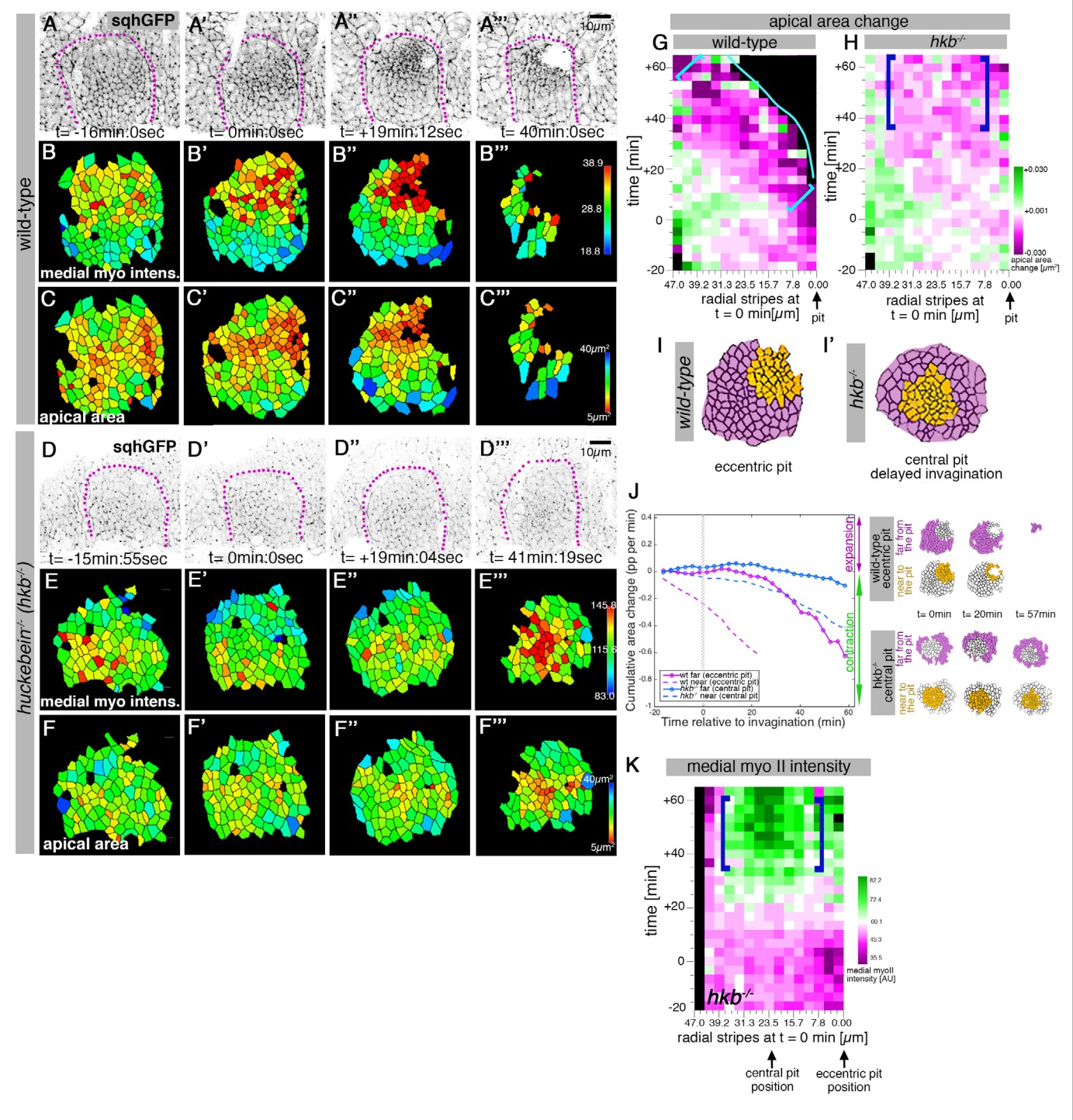

**Figure 3.** *hkb⁻/⁻* mutants show a delayed symmetrical apical constriction. (**A-F‴**) Stills of a representative wild-type (**A-C‴**) and a *hkb⁻/⁻* mutant (**D-F‴**) embryo time lapse movie at the indicated time points. (**A-A‴**) and (**D-D‴**) show the SqhGFP channel of the movies visualising myosin II, (**B-B‴**) and (**E-E‴**) show the average apical-medial myosin II intensity of segmented placodal cells, (**C-C″**) and (**F-F‴**) show the corresponding apical cell areas. (**E-E‴**) *hkb⁻/⁻* mutants show apical-medial accumulation of myosin II at the centre at a delayed time point. Indicated colour scales correspond to the 5-95% range in each movie. Magenta dotted lines indicate the boundary of the placode. (**F-F‴**) The delayed accumulation of myosin II in *hkb⁻/⁻* mutants is mirrored by the changed apical constriction pattern, with the wild-type constricting in the dorsal-posterior corner (**C-C‴**) and *hkb⁻/⁻* mutants in the centre, but very delayed (**F-F‴**). (**G, H**) Rate of apical area change of cells in wild-type (**G**; data from 7 movies) and *hkb⁻/⁻* mutant (H; data from 7 movies) embryos shown over time and across radial distance of cells from the eccentric wild-type pit location (0 µm) at t = 0 min. 47 µm represents the boundary of the placode. The blue line in (**G**) indicates where cells have just disappeared into the pit in the wild-type. Greatest apical area change in the wild-

*Figure 3 continued on next page*

*Figure 3 continued*

type is always confined to the area in front of the pit (brackets in **G**), whereas in *hkb*⁻/⁻ mutant embryos cells in a broader central region show a delayed apical constriction (blue brackets in **H**). See also *Figure 3—figure supplement 1*. (**I, I'**) Schematic representation of a segmented wild-type and *hkb*⁻/⁻ mutant placode, with cells near the eccentric wild-type pit and central *hkb*⁻/⁻ mutant pit/area of constriction indicated in orange, and cells far from the pit for both in magenta. (**J**) Comparative analysis of cumulative apical area change of *hkb*⁻/⁻ mutant embryos expressed as proportion per minute (pp/min), pooled from 7 time-lapse movies for cells that were specified at t = 0 min to be located near the central constricting pit (dashed blue curve; see schematic on the right) and cells that were specified at t = 0 min to be located far from the central pit (solid blue line; see schematic on the right) versus wild-type placodal cells as previously shown in *Figure 1C*. (**K**) Spatial representation of the average medial myosin intensity from an exemplary time-lapse movie of a *hkb*⁻/⁻ mutant embryo, with radial location (collapsed into stripes) of cells specified at t = 0min. The blue brackets mark the peak of medial myosin intensity that is located in the centre of the placode in *hkb*⁻/⁻ mutant embryos rather than tracking near the forming eccentric pit as in the wild-type. In all analyses of the wild-type, t = 0min is defined as the frame just before the first sign of invagination at the future pit was evident. *hkb*⁻/⁻ mutants were aligned using as a reference of embryo development the level of invagination of the tracheal pits that are not affected in the *hkb*⁻/⁻ mutant as well as other morphological markers such as appearance and depth of segmental grooves in the embryo. See also *Figure 3—figure supplements 1 and 2* and *Figure 3—videos 1–4*.

The online version of this article includes the following video and figure supplement(s) for figure 3:

**Figure supplement 1.** *hkb*⁻/⁻ mutants show a delayed symmetrical apical constriction.

**Figure supplement 2.** *hkb*⁻/⁻ mutants show a delayed symmetrical apical constriction followed by aberrant invagination.

**Figure 3—video 1.** Extended wild-type embryo movie 1 for strain rate analysis.

https://elifesciences.org/articles/72369/figures#fig3video1

**Figure 3—video 2.** Extended wild-type embryo movie 2 for strain rate analysis.

https://elifesciences.org/articles/72369/figures#fig3video2

**Figure 3—video 3.** Extended *hkb*⁻/⁻ mutant embryo movie 1 for strain rate analysis.

https://elifesciences.org/articles/72369/figures#fig3video3

**Figure 3—video 4.** Extended *hkb*⁻/⁻ mutant embryo movie 2 for strain rate analysis.

https://elifesciences.org/articles/72369/figures#fig3video4

had occurred yet (*Figure 3D''', E''' and F'''*). Eventually, these embryos invaginated a dilated tube through an enlarged pit (see *Figure 3—figure supplement 2*). We analyzed the spatial distribution of apical area changes over time in wild-type and *hkb*⁻/⁻ placodes (*Figure 3G and H*). In the wild-type, the area of strongest apical area constriction was always located in a position near the pit, similar to the distribution of medial myosin intensity and activity (*Figure 2D and E*). In contrast, in *hkb*⁻/⁻ mutant placodes cells began to constrict in an aberrant central position from about t=+20–30 min (*Figure 3G and H*). Thus, *hkb*⁻/⁻ mutants show a delayed central invagination pit compared to the eccentric pit observed in wild-type placodes (*Figure 3(I)1'*). For our further quantitative analyses, we now took this change to a central pit position in the mutants into account. We plotted cumulative apical area change for cells in *hkb*⁻/⁻ mutant placodes, split into cells near the (now central) pit and cells far from the pit (at the placode periphery). This analysis showed that cells in *hkb*⁻/⁻ mutant placodes located near the central pit started to constrict at a much slower rate than cells in wild-type placodes, and only from about t=+20 min on (*Figure 3J*, wild-type data reproduced from *Figure 1E*). In fact, the rate of the constriction of the central cells in *hkb*⁻/⁻ mutants was very similar to that of the cells located at a distance to the pit in the wild-type (compare blue dashed and magenta solid curves), suggesting that the central cells in *hkb*⁻/⁻ mutants continue to behave identical to the wild-type cells occupying the same position. Cells far from the central pit in *hkb*⁻/⁻ mutants showed a slight apical area expansion up to t=+20 min followed by a slight constriction, but did not invaginate over the time interval shown in the plots. In agreement with the delayed central constriction of cells in *hkb*⁻/⁻ mutant placodes, we found no accumulation of apical-medial actomyosin at the eccentric position where a pit forms in the wild-type (compare *Figure 3E and K* to *Figure 2D*). Instead, only at t~+40 min did a central group of cells in *hkb*⁻/⁻ mutant placodes display increased accumulation of apical-medial myosin II (*Figure 3K*), inducing a shallow tissue bending in this region at this point (see also *Figure 3—figure supplement 2*).

Thus, in comparison to wild-type placode with an eccentric invagination, in *hkb*⁻/⁻ mutant placodes a central invagination formed with a significant delay in time.

## *hkb*⁻/⁻ mutants display disrupted cell behaviors across the placode

The above analyses strongly suggest that the initial asymmetrical setup of cell behaviors in the placode might be disrupted in *hkb*⁻/⁻ embryos. We therefore performed a strain (deformation) rate analysis

of placodes in $hkb^{-/-}$ mutants in comparison to wild-type embryos to assess this quantitatively. The strain rate analysis is based on the assumption that any change in tissue shape can be accounted for by changes in cell shape and cell intercalations (*Blanchard et al., 2009*; *Sanchez-Corrales et al., 2018*). In many tissues, cell division or cell death and delamination also contribute to strain rates, but these are absent in the salivary gland placode during invagination and can thus be neglected. The origin of the radial coordinate system used for this analysis was specified at t=0 min and positioned at the center of of the placode in the $hkb^{-/-}$ mutant embryos (where the central constriction eventually occurred) and in wild-type embryos at the dorsal-posterior corner of the placode (where the wild-type pit will form). In wild-type placodes, cells that were located near the eccentric invagination pit (*Figure 4A*; with the 'near' position defined at t=0 min; see also *Figure 4—figure supplement 1*) constricted isotropically, with near equal change in strain at the tissue and cell shape level, and very little intercalation strain (*Figure 4A' and A''*, solid curves). These cells in wild-type placodes could only be tracked up to t=20 min as at this point they had all invaginated into the embryo to form the initial part of the tube. In contrast, in the $hkb^{-/-}$ mutants, cells near the forming central pit (*Figure 4A*; defined to be in a 'near' position at t=0 min) showed a much reduced tissue and cell shape strain rate, with near identical radial and circumferential contributions, and constriction only commenced at about half the rate of the wild-type placodes from t=+20 min onwards (*Figure 4A' and A''*, dashed curves; see also *Figure 4—figure supplement 1*). In wild-type placodes, cells far from the eccentric pit (*Figure 4B*; defined to be 'far' at t=0 min), showed cell intercalations leading to circumferential convergence and radial extension of the tissue over the first approximately 20 min after invagination commenced, as we also reported previously (*Figure 4B' and B''*, solid curves; tissue and intercalation strain rate showing expansion radially and contraction circumferentially; *Sanchez-Corrales et al., 2018*). Interestingly, we then observed a clear switch in cell behavior of these cells to a second phase after t=+20 min. Once these cells were located closer to the pit position, they now also displayed apical contraction (beyond t=+20 min; *Figure 4B' and B''*, solid green curves). In the $hkb^{-/-}$ mutants, for cells at the placode periphery and hence far from the central constriction (*Figure 4B*), the tissue expanded slightly in both radial and circumferential direction (*Figure 4B' and B''*, dashed gray curves), with cell shapes narrowing radially (dashed green curve in *Figure 4B'*) and extending circumferentially (dashed green curve in *Figure 4B''*). We could observe a similar rate of intercalation in these cells as in cells far from the pit in wild-type placodes, with some radial elongation and circumferential contraction (dashed orange curves in *Figure 4B' and B''*). As the tissue strain rate in the $hkb^{-/-}$ mutants showed no overall signature of directional changes with only a slight expansion both radially and circumferentially, we were curious to understand better what these tissue-scale signatures could represent at the individual cell level.

We therefore analyzed individual events of neighbor gains across the tissue in wild-type and $hkb^{-/-}$ mutant embryos. We compared two types of intercalation events, those leading to circumferential neighbor gains that in wild-type placodes explain the observed intercalation strain rates, and those leading to radial neighbor gains, and thus potentially opposing the changes seen at the tissue and intercalation strain rate level (*Figure 4C*). In wild-type placodes, circumferential neighbor gains outweigh radial ones by a large margin throughout (*Figure 4D*, dashed curves), leading to a steady increase in cumulative productive neighbor gains for cells across the placode (*Figure 4E*, yellow curves). In contrast, in $hkb^{-/-}$ mutant embryos circumferential and radial neighbor gains across the placode were overall much reduced in number and occurred with equal frequency (*Figure 4D*, solid curves). Furthermore, cells near the delayed central pit in $hkb^{-/-}$ mutants showed no productive neighbor gains (defined as the difference between circumferential and radial neighbor gains, corrected for the proportion of interfaces that could successfully be tracked; *Figure 4E*, purple dashed curve), whereas cells at the periphery away from the central pit showed productive neighbor gains (*Figure 4E*, solid purple curve). Active neighbor gains in the placode and other morphogenetic events are initiated and driven by junctional actomyosin (*Sanchez-Corrales et al., 2018*; *Tetley et al., 2016*), prompting us to analyze junctional myosin polarisation in $hkb^{-/-}$ mutant embryos. We previously reported on the unipolar and bipolar enrichment of junctional myosin in the wild-type at early stages of placode morphogenesis (*Sanchez-Corrales et al., 2018*) and expand it here to the whole process of secretory cell invagination in comparison to $hkb^{-/-}$ mutants (*Figure 4F and G*). Whereas in control placodes junctional myosin is clearly enriched in circumferential junctions by both measures, up to the point of most secretory cells having internalized by t=+40 min (*Figure 4F and G*; dashed green curve vs. dashed purple curve),

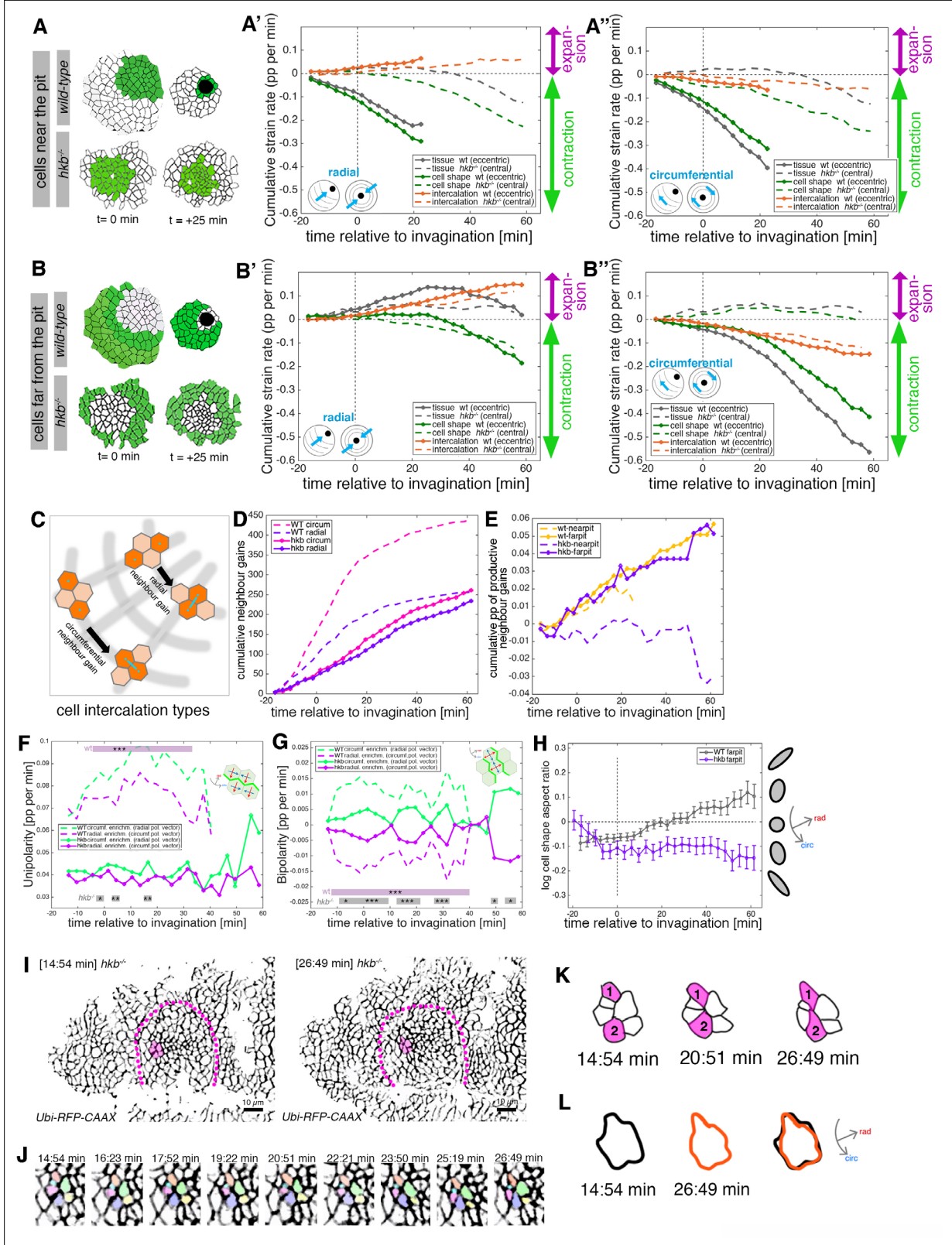

Figure 4. *hkb*[−/−] mutants show aberrant cell behaviors during invagination. (**A–B"**) Regional breakdown of time-resolved cumulative strain rates, with regions defined at t=0 min based on eccentric pit for the wild-type and a central pit for *hkb*[−/−] mutants. For cells 'near the pit' in the wild-type (**A**), tissue constriction dominates (solid gray curves in (**A'**, **A"**)) and is due to isotropic cell constriction (solid green curves in (**A'**, **A"**)), whilst intercalation only plays a minor role in this region (solid orange curves in (**A'**, **A"**)). Cells in this region have completely internalized by about t=20 min. By contrast, in *hkb*[−/−]

*Figure 4 continued on next page*

*Figure 4 continued*

mutant embryos, cells 'near the (central) pit' (**A**), show strongly reduced tissue (dashed gray curves in (**A′, A″**)) and cell strain rates (dashed green curves in (**A′, A″**)) and mildly reduced intercalation (dashed orange curves in (**A′, A″**)). For cells 'far from the pit' in the wild-type (**B**), the tissue elongates toward the pit until t=20 min (solid gray curve in (**B′**)), with a corresponding contraction circumferentially (solid gray curve in (**B″**)), and this is predominantly due to cell intercalation (solid orange curves in (**B′, B″**)). Beyond t=20 min, these cells have reached the invagination pit and also constrict isotropically, thereby leading the tissue change (solid gray curves in (**B′, B″**) >20 min) to mirror the cell shape change (solid green curves in (**B′, B″**) >20 min). By contrast, in *hkb*⁻/⁻ mutant embryos, cells 'far from the (central) pit' (**B**) show a slight tissue expansion both radially and circumferentially (dashed gray curves in (**B′, B″**)), paired with abnormal circumferential cell elongation (dashed green curves in (**B′, B″**)), and some reduced intercalation (dashed orange curves in (**B′, B″**)). The corresponding instantaneous strain rate plots can be found in *Figure 4—figure supplement 1*. Data from nine wild-type movies and five *hkb*⁻/⁻ movies were analyzed (see *Figure 3—figure supplement 1*). (**C–E**) Quantification of neighbor gains as a measure of T1 and intercalation events. Examples of a circumferential neighbor gain (leading to radial tissue expansion), and a radial neighbor gain (leading to circumferential tissue expansion) are shown in (**C**). (**D**) Circumferential neighbor gains dominate over radial neighbor gains in the wild-type (dashed curves), with the rate of neighbor exchanges dropping beyond 20 min. In contrast, in *hkb*⁻/⁻ mutant embryos, the amount of circumferential and radial gains is identical (solid curves). (**E**) Cumulative proportion of productive neighbor gains, defined as the amount of circumferential neighbor gains leading to radial tissue elongation and expressed as a proportion (pp) of cell-cell interfaces tracked at each time point, and split into cells near the pit (eccentric for wild-type and central for *hkb*⁻/⁻ mutant) and far from the pit. Predicted productive neighbor gains are strongly reduced and near zero for cells near the pit in *hkb*⁻/⁻ mutants compared to control (dashed curves), whereas cells far from the pit in *hkb*⁻/⁻ mutant continue to intercalate similar to wild-type (solid curves). (**F, G**) Myosin II junctional polarity was quantified from segmented and tracked time-lapse movies. Myosin enrichment at junctions can occur in two flavors: Myosin II unipolarity is defined as myosin II enrichment selectively on side of a cell ((**F**), see schematic inset). Myosin II bi-polarity is defined as myosin II enrichment at two parallel oriented junctions of a single cell, calculated as the magnitude of a vector pointing at the enrichment (**G**). Data from six wild-type movies and five *hkb*⁻/⁻ movies, number of cells is shown in *Figure 3—figure supplement 1*. Plotted are the rates of change of the uni- and bipolarity amplitudes as a proportion per minute (pp/min) of the mean cell perimeter fluorescence. (**F**) Circumferential myosin II uni-polar enrichment (i.e., the radial uni-polarity vector, red arrow in schematic, pointing at the myosin enrichment), increases and is high until ~40 min when it drops ((**G**), green dashed curve). The circumferential uni-polar enrichment is always higher than the radial myosin II uni-polar enrichment (green solid curve in (**G**)). The myosin II uni-polar enrichment in *hkb*⁻/⁻ mutants is overall strongly reduced compared to wild-type (solid curves in (**F**)). (**G**) Circumferential myosin II bi-polar enrichment in the wild-type (i.e., the radial bi-polarity vector, red arrow in schematic, pointing at the myosin enrichment) is high until ~40 min when it drops ((**G**), green dashed curve). Until this point, it is higher than the radial myosin II bi-polar enrichment (green solid curve in (**G**)). The myosin II bi-polar enrichment in *hkb*⁻/⁻ mutants is strongly reduced compared to the wild-type (solid curves in (**G**)). Statistical significance of $p < 0.05$ (*), $p < 0.005$ (**), $p < 0.0005$ (***) using a mixed-effect model is indicated as shaded boxes at the top and bottom of the panels: comparing circumferential over radial enrichment for either the wild-type or *hkb*⁻/⁻ mutants. (**H**) Analysis of cell shape aspect ratio dynamics in cells far from the pit (eccentric pit for wild-type, cental pit for *hkb*⁻/⁻ mutant). In the wild-type, circumferential elongation as part of active circumferential neighbor gains (*Sanchez-Corrales et al., 2018*) persist until ~t=+20 min, when cells start to become elongated radially (gray curve). In *hkb*⁻/⁻ mutants, cells become and remain circumferentially elongated (magenta curve). Data shown for seven wild-type and five *hkb*⁻/⁻ mutant movies. (**I–L**) Analysis of an exemplary cell intercalation event in cells far from the pit in a segmented and tracked time-lapse movie of a *hkb*⁻/⁻ mutant placode, stills at the beginning and end of the event shown are in (**I**), and stills of the whole event in (**J**). (**K**) Cells 1 and 2 gain a circumferential contact, mainly via cell elongation. (**L**) The cell cluster is already circumferentially elongated at t=+14:54 min (black outline) and remains near identically elongated at t=+26:49 min (orange outline). Data for (**A–H**) pooled from nine wild-type and five *hkb*⁻/⁻ mutant movies. In all analyses of the wild-type, t=0 min is defined as the frame just before the first sign of invagination at the future pit was evident. *hkb*⁻/⁻ mutants were aligned using as a reference of embryo development the level of invagination of the tracheal pits that are not affected in the *hkb*⁻/⁻ mutant as well as other morphological markers such as appearance and depth of segmental grooves in the embryo. Panels (**A, A′, B, B′, F, G**) are expressed as proportion per minute (pp/min).

The online version of this article includes the following figure supplement(s) for figure 4:

**Figure supplement 1.** *hkb*⁻/⁻ mutants show aberrant cell behaviors.

---

in *hkb*⁻/⁻ mutant placodes myosin unipolarity and bipolarity are strongly reduced, even though some bipolarity remained (*Figure 4F and G*, solid curves). Thus, the major driver of active intercalations appeared to be strongly diminished in the mutants, posing the question of what could drive the observed intercalations here. We also analyzed cell elongation in cells far from the pit, with cells in wild-type placodes being initially elongated circumferentially at t<+20 min, while early intercalation events are initiated by circumferential junctional myosin (*Figure 4H*, gray curve; *Sanchez-Corrales et al., 2018*). These cells then became increasingly elongated radially as they moved into a position nearer to the pit and were being actively funneled toward the pit. In *hkb*⁻/⁻ mutant embryos, cells far from the pit at the periphery of the placode very quickly became circumferentially elongated and remained so throughout the time period analyzed. This aberrant circumferential elongation together with the mild bipolarity of junctional myosin polarisation that we observed prompted us to analyze individual events of cell intercalation in time-lapse movies (*Figure 4I–L*). In cells far from the central pit in *hkb*⁻/⁻ mutants, we could observe groups of cells clearly undergoing intercalations (*Figure 4I and J*) with individual cells making circumferential neighbor gains and elongating (cells 1 and 2 in *Figure 4K*), whilst the overall shape of the local group of cells undergoing the intercalation event was

not changing (*Figure 4L*). Thus, although cells far from the aberrant central pit in *hkb*$^{-/-}$ mutants still intercalated and gained circumferential neighbors, this behavior lacked several key signatures of wild-type intercalations, and overall in the *hkb*$^{-/-}$ mutants intercalations did not lead to convergence and extension of the tissue toward the misplaced pit.

These analyses show that in the absence of Hkb, the initial radial patterning of cell behaviors across the placode is highly aberrant. Cells located at the position corresponding to where the eccentric pit would be located in the wild-type clearly failed to constrict apically, suggesting that Hkb could be involved in the specification of this pit behavior observed in wild-type.

## The salivary gland placodal primordium is asymmetrically patterned by Hkb and Fkh prior to morphogenesis

The loss of correct patterning of cell behaviors in the early placode in *hkb*$^{-/-}$ mutant embryos suggested that Hkb was involved in establishing these in wild-type embryos. Furthermore, *hkb* mRNA expression has been reported to show a dynamic pattern by in situ hybridization and β-gal reporter labeling (*Myat and Andrew, 2000b*). In order to be able to analyze Hkb protein levels dynamically and in comparison to cell shapes, we generated a Venus-tagged version of Hkb using CRISPR/Cas9 to tag the endogenous protein. Using this in vivo reporter, we detected Venus-Hkb at high levels concentrated in the dorsal-posterior region of the just forming placode, in the area where the invagination pit will form, already at t=–30 min prior to the first tissue bending (*Figure 5A, A', D and M* and *Figure 5—figure supplement 1B,B''* ), and even as early as t=–63 min prior to invagination (*Figure 5—figure supplement 1A*). The expression then expanded more broadly and anteriorly across the whole placode over time (*Figure 5B–C', E, F and M* and *Figure 5—video 1*). At the moment of first tissue bending (t=0 min) Venus-Hkb was found across the placode, with elevated levels still at the invagination pit position, and a second increase toward the anterior. At late stages (t>>0 min) Venus-Hkb levels close to the invagination point were lower than further anterior (*Figure 5M*).

As mentioned above, embryos mutant in *fkh* also show a lack of early apical constriction at the dorsal-posterior corner of the placode (*Myat and Andrew, 2000a*; *Sanchez-Corrales et al., 2018*). Similar to Hkb, analysis of *fkh* mRNA and protein levels in fixed embryos indicated an early enrichment near the forming invagination pit (*Myat and Andrew, 2000a*; *Sanchez-Corrales et al., 2018*). Using an in vivo genomic GFP-Fkh reporter (*Spokony and White, 2013*), we followed Fkh protein levels and patterning over the same time period whilst also assessing apical cell shapes (*Figure 5G–L*). Already at t=–60 min prior to tissue bending, Fkh-GFP was visible in the cells near the future invagination pit (*Figure 5G, G', J and N* and *Figure 5—figure supplement 1C–C''*), from where it spread anteriorly and ventrally across the whole placode (*Figure 5—video 2*). At t=0 min, Fkh-GFP was already detected in all cells of the placode, though with levels still highest near the pit and decreasing toward the anterior edge of the placode (*Figure 5H, H', K and N*). At t=+30 min and beyond, levels of Fkh-GFP were more uniform across the placode (*Figure 5I, I', L, N*).

Thus, both Hkb and Fkh transcription factors showed a clear temporally and spatially graded pattern of expression within the early placode that developed dynamically. Changes were especially pronounced at the dorsal-posterior corner, with highest levels here already an hour before tissue bending commenced. Combined Hkb and Fkh action could thereby define the future position of the invagination pit. We therefore compared Venus-Hkb as well as Fkh-GFP fluorescence intensity levels from time-lapse movies that in parallel allowed us to determine the corresponding apical area of cells (*Figure 5O–P'*). Venus-Hkb intensity levels peaked in cells located at the site of the forming eccentric invagination pit at about t ~ –30 min (*Figure 5O*), with an appearance of smaller apical areas in this position following on about 30–40 min later (*Figure 5O'*). Similarly, Fkh-GFP intensity began to peak at about the same time, t ~ –30 min, and then tracked at high level at a fixed distance to the pit over time (*Figure 5P*). Again also here, the increased reduction in apical area appeared with a delay of about 30–40 min (*Figure 5P'*). Such a delay would be consistent with the estimated time required for the transcription of the *fog* locus, followed by translation, protein folding, secretion, binding to the receptor, and signaling leading to myosin phosphorylation (*Garcia et al., 2013*; *Shamir et al., 2016*).

Thus, Hkb and Fkh dynamic expression at the future eccentric pit could prefigure the start of apical constriction at this site. The continuous high levels of Fkh tracking at a fixed distance to the pit suggest that Fkh is key to the continuous apical constriction of cells moving into a near pit position. These dynamic changes might trigger the graded and patterned expression of targets of both

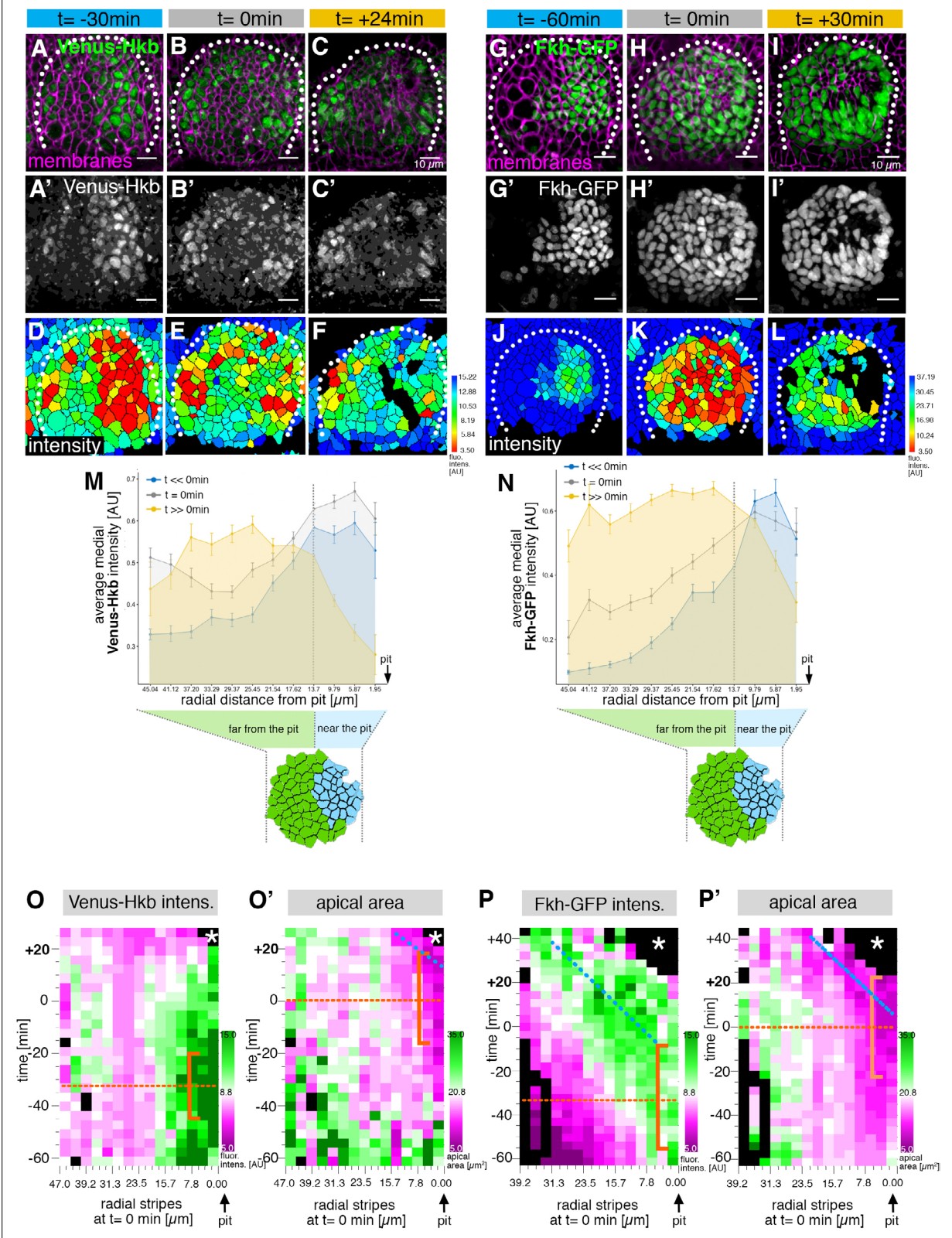

**Figure 5.** The salivary gland placodal primordium is asymmetrically patterned by Hkb and Fkh prior to morphogenesis. (**A–F**) Stills of a time-lapse movie of embryos with endogenously tagged Hkb, Venus-Hkb, at t=–38 min, +0 min and +38 min show the dynamic pattern of expression of Hkb. Expression starts at the posterior corner before tissue bending commences. Cell membranes are in magenta in (**A–C**), Venus-Hkb is green in (**A–C**) and as a single channel in (**A'–C'**). (**D–F**) show the quantification of Venus-Hkb fluorescence intensity at these time points. (**G–L**) Stills of a time-lapse movie

*Figure 5 continued on next page*

*Figure 5 continued*

of embryos with Fkh tagged by GFP under endogenous expression control, at t=–60 min, +0 min and +30 min show that as early as 1 hr before tissue bending, Fkh is already expressed at the posterior corner. Cell membranes are in magenta in (**G–I**), Fkh-GFP is green in (**G–I**) and as a single channel in (**G'–I'**). (**J–L**) show the quantification of Fkh-GFP fluorescence intensity at these time points. (**M**) Venus-Hkb expression starts at the posterior corner in a region similar to 'near the pit' population (between pit location and dashed line). At the start of invagination (around t=0 min) the expression increases across the placode, in particular at the anterior edge. At late stages, the level of Venus-Hkb expression decreases in the pit region, but it remains high in the rest of the placode. Data is pooled from selected frames of time-lapse movies and corresponding fixed samples. Number of cells are as follows: t<<0 min (equivalent to stage 10), n=777 cells from eight embryos; t=0 min (equivalent to early stage 11), n=1186 cells from 10 embryos and t>>0 min (equivalent to late stage 11/early stage 12), n=885 cells from 10 embryos. (**N**) Fkh-GFP expression initiates at the posterior corner and increases to cover the whole placode over time. Data is pooled from selected frames of time-lapse movies and corresponding fixed samples. Number of cells are as follows: t<<0 min (equivalent to stage 10), n=365 cells from four embryos; t=0 min (equivalent to early stage 11), n=1026 cells from nine embryos and t>>0 min (equivalent to late stage 11/early stage 12), n=659 cells from nine embryos. (**O, O'**) Spatial representation of the Venus-Hkb intensity (**O**) and apical area (**O'**) from an exemplary time-lapse movie of a wild-type embryo, with radial location (collapsed into stripes) of cells specified at t=0 min. The orange bracket in (**O**) marks the appearing peak of high Venus-Hkb intensity at the position of the eccentric pit, with the orange dotted line marking the highest point of intensity. The orange bracket in (**O'**) marks the appearing apical constriction at the position of the eccentric pit, with the orange dotted line marking the midway bracket point. The blue dotted line in (**O'**) marks where the smallest apical area begins to track in the near pit position (as shown in *Figure 3G*). (**P, P'**) Spatial representation of the Fkh-GFP intensity (**P**) and apical area (**P'**) from an exemplary time-lapse movie of a wild-type embryo, with radial location (collapsed into stripes) of cells specified at t=0 min. The orange bracket in (**P**) marks the appearing peak of high Fkh-GFP intensity at the position of the eccentric pit, with the orange dotted line marking the highest point of intensity. The orange bracket in (**P'**) marks the appearing strongest apical constriction at the position of the eccentric pit, with the orange dotted line marking the midway bracket point. The blue dotted lines in (**P**) and (**P'**) mark where the Fkh-GFP intensity and the smallest apical area begin to track in the near pit position. In all analyses, t=0 min is defined as the frame just before the first sign of invagination at the future pit was evident. See also *Figure 5—figure supplement 1*.

The online version of this article includes the following video and figure supplement(s) for figure 5:

**Figure supplement 1.** The salivary gland placodal primordium is asymmetrically patterned by Hkb and Fkh prior to morphogenesis.

**Figure 5—video 1.** Hkb dynamics in the placode.

https://elifesciences.org/articles/72369/figures#fig5video1

**Figure 5—video 2.** Fkh dynamics in the placode.

https://elifesciences.org/articles/72369/figures#fig5video2

transcription factors, leading to the observed patterned cell behaviors we uncovered, as well as the switch between intercalation and constriction.

## Asymmetric Fog expression is controlled by Hkb and Fkh and is upstream of early differential behaviors in the placode

With the transcription factors Fkh and Hkb both displaying intriguing protein expression patterns across the early placode pre-morphogenesis, targets of both factors are likely to play key roles in instructing cell behaviors such as wedging and cell intercalation. Interestingly, an upstream activator of apical-medial actomyosin activity, the GPCR-ligand Folded gastrulation (Fog) was found to be dependent on Fkh (*Chung et al., 2017*; *Dawes-Hoang et al., 2005*; *Figure 6*) and suggested to be downstream of Hkb (*Myat and Andrew, 2000b*; though in this publication as 'data not shown'). Fog is an apically secreted ligand that acts in an autocrine fashion, and thus the cells that switch on Fog expression will apically constrict (*Dawes-Hoang et al., 2005*). Previously published in situ hybridization of *fog* mRNA suggested increased levels near the forming pit, similar to the early Hkb and Fkh enrichment (*Nikolaidou and Barrett, 2004*). Fog expression and signaling in particular could not only be upstream of the isotropic constriction of cells near the forming pit, but could also be involved in maintaining this behavior in new coronas of cells moving into a position near the pit over time.

We therefore decided to carefully analyze Fog protein levels in comparison to apical area and apical-medial myosin intensity in salivary gland placodes from early to late invagination stages (*Figure 6A–I*). Fog levels were low at mid-stage 10 concomitant with placode specification (*Figure 6A, A' and C*). Just before tissue bending commenced at t=0 min, Fog was already strongly enriched in cells near the forming invagination pit, and these cells were the ones already showing the decreased apical area (*Figure 6D–F*). Fog protein levels remained enriched near the invagination point once the tube had started to internalize (*Figure 6G–I*). In fact, Fog protein levels in cells were positively correlated with both smaller apical areas (*Figure 6K*) as well as higher average apical-medial myosin II intensity (*Figure 6L*; *Figure 6—figure supplement 1*).

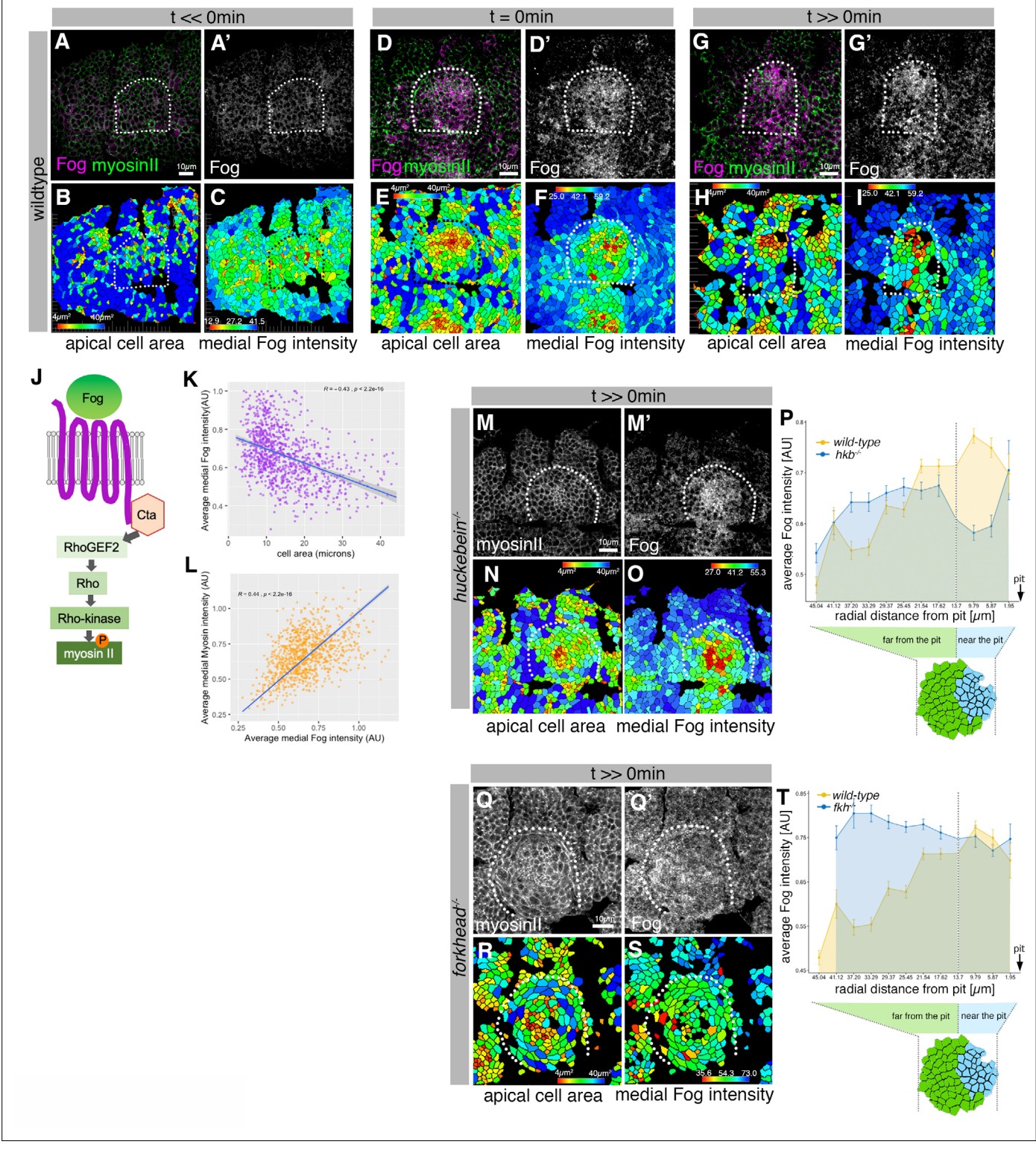

**Figure 6.** Asymmetric Fog expression is controlled by Hkb and Fkh upstream of early differential behaviors in the placode. (**A–I**) Wild-type embryos labeled for Fog protein (the ligand to the GPCR upstream of Rho-dependent myosin II activation, (**J**)) and myosin II, visualized by SqhGFP. Fog is magenta in (**A, D, G**) and as a single channel in (**A', D', G'**), myosin II is green in (**A, D, G**). Also shown are corresponding quantifications of apical cell area (**B, E, H**) and average medial Fog fluorescence intensity (**C, F, I**). Time points analyzed were before any apical constriction and tissue bending commencing (t<<0 min; ~stage 10), at t=0 min (~early stage 11) and once invagination had commenced (t>>0 min; ~late stage 11/early stage 12). (**J**) Schematic of the GPCR pathway leading to myosin II activation. (**K**) Average medial Fog fluorescence intensity negatively correlates with apical cell area.

*Figure 6 continued on next page*

*Figure 6 continued*

Pearson coefficient r=–0.436; n=823 cells from seven embryos. (**L**) Average medial myosin II fluorescence intensity positively correlates with average medial Fog fluorescence intensity. Pearson coefficient r=0.439; n=823 cells from seven embryos. (**M–O**) In *hkb*$^{-/-}$ mutant embryos at a time point where invagination would have been well advanced in wild-type embryos (t>>0 min), medial myosin II (**M**) and Fog fluorescence ((**M'**) and quantified in (**O**)) are concentrated in cells in the center of the placode, where cells also show constricted apices (**N**). (**P**) Quantification of average medial Fog fluorescence intensity according to radial position in wild-type versus *hkb*$^{-/-}$ mutant embryos. Whereas in the wild-type Fog intensity peaks at the posterior end of the placode where the invagination pit forms, in *hkb*$^{-/-}$ mutant embryos Fog is dramatically reduced at the invagination pit. Number of cells: wild-type n=823 from seven embryos; *hkb*$^{-/-}$ n=765 cells from five embryos. (**Q–S**) In *fkh*$^{-/-}$ mutant embryos at a time point where invagination would have well advanced in wild-type embryos (t>>0 min), medial myosin II (**Q**) and Fog fluorescence ((**Q'**) and quantified in (**S**)) are very homogenous across the placode and no longer enriched within the placode compared to the surrounding epidermis. Apical area quantification shows there is only a mild central cell constriction and no invagination (**R**). (**T**) Quantification of average medial Fog intensity in *fkh*$^{-/-}$ mutant embryos. In contrast to wild-type embryos with Fog enrichment at the posterior pit, levels of Fog are more homogenous in the placode in *fkh*$^{-/-}$ mutant embryos with an increase toward the anterior. Levels are comparable to the rest of the epidermis (see *Figure 3—video 3*). The intensity curves shown for wild-type and *fkh*$^{-/-}$ mutant embryos are comparable in their shape, though the absolute intensities shown are not directly comparable, as fluorescence intensity was normalized in each image by dividing the average fluorescence per cell by the 98th percentile value to account for embryo to embryo variability in staining efficiency. Number of cells: wild-type n=823 from seven embryos; *fkh*$^{-/-}$ mutant n=512 cells from five embryos. In all analyses of the wild-type, t=0 min is defined as the frame just before the first sign of invagination at the future pit was evident. *hkb*$^{-/-}$ mutants were aligned using as a reference of embryo development the level of invagination of the tracheal pits that are not affected in the *hkb*$^{-/-}$ mutant as well as other morphological markers such as appearance and depth of segmental grooves in the embryo. See also *Figure 6—figure supplement 1*.

The online version of this article includes the following figure supplement(s) for figure 6:

**Figure supplement 1.** Asymmetric Fog expression is controlled by Hkb and Fkh and is upstream of early differential behaviors in the placode.

*hkb*$^{-/-}$ mutant embryos, as detailed above, showed a very clear disruption of the early asymmetric placodal patterning. At late stage 11, Fog protein levels in *hkb*$^{-/-}$ mutant embryos appeared to be similar to wild-type embryos and enriched in the placode compared to the surrounding epidermis, though the pattern of Fog enrichment was altered. Cells at the posterior corner, where the invagination pit would have formed in wild-type placodes, lacked high Fog protein levels, whereas only cells in the center of the placode now showed high levels of Fog protein (*Figure 6M–O*). When quantitatively analyzing many *hkb*$^{-/-}$ mutant embryos in comparison to wild-type, the peak intensity of Fog protein was clearly shifted to a central, much more symmetrical, position (*Figure 6P*). In contrast, although we could detect some Fog protein in *fkh*$^{-/-}$ embryos at late stage 11, it was no longer enriched at the position where the invagination pit forms in the wild-type. The levels also did not appear raised anywhere within the placode compared to the surrounding epidermis (*Figure 6Q-S* and *Figure 6— figure supplement 1* D-D''), as the quantification of many placodes confirmed (*Figure 6T*). In both mutants, altered levels and patterning of Fog protein still correlated with the now altered levels and patterns of residual apical-medial myosin (*Figure 6M and Q*).

In order to understand the cause of the central and delayed constriction in *hkb*$^{-/-}$ mutant embryos upstream of the also only centrally localized Fog, we examined Fkh protein localization in *hkb*$^{-/-}$ mutants (*Figure 6—figure supplement 1E-J*). At late stage 10, when the Fkh protein expression pattern is just expanding from the eccentric pit position to the central part of the placode in the control, the pattern was indistinguishable in the *hkb*$^{-/-}$ mutants (*Figure 6—figure supplement 1* E-F' and H-I'). The pattern remained identical to wild-type placodes throughout stage 11 and beyond (*Figure 6—figure supplement 1* G,G', J, J'). Hence the natural expansion of Fkh protein expression to the cells in the center of the placode that is also observed in the *hkb*$^{-/-}$ mutant placodes might be the cause of the delayed constriction of the cells in this region, whereas the earlier initial constriction at the site of the eccentric pit was absent in the mutant because it requires both Fkh and Hkb. Both transcription factors appeared in fact to be independently regulated, as also Hkb levels and protein expression pattern were unchanged in *fkh*$^{-/-}$ mutant embryos (*Figure 6—figure supplement 1* K-N').

Thus, the asymmetrical setup of the salivary gland placode prior to initiation of tube budding relies on a correct spatial and temporal patterning of the key transcription factors, Hkb and Fkh. These transcription factors are in turn upstream of GPCR/Fog signaling that leads to patterned and sustained apical-medial myosin activity and hence contractile cell behaviors.

## Changes to invagination pit size or position impact invagination and perturb final organ shape

Our data and previously published results show that the asymmetrical invagination point depends on patterning by transcription factors and downstream GPCRs/Fog signaling at the posterior corner, leading to a focussed and point-like invagination pit (*Chung et al., 2017*; *Myat and Andrew, 2000a*; *Myat and Andrew, 2000b*; *Myat and Andrew, 2002*; *Sanchez-Corrales et al., 2018*). We next sought to test the importance of this asymmetric setup for successful morphogenesis of a narrow-lumen and symmetrical tube. To do so, we overexpressed Fog in the salivary gland placode only, using *UAS-Fog x fkhGal4* which led to placode-wide increased levels of Fog from early stages on (*Figure 7* and *Figure 7—figure supplement 1*). Compared to the focused apical constriction near the forming invagination pit in wild-type placodes at early stage 11 (*Figure 7A and A'*), when Fog was overexpressed across the placode most secretory cells of the placode constricted simultaneously (*Figure 7B, B', C and C'*). At late stage 11, when in wild-type placodes cells had begun to invaginate and a short symmetrical narrow-lumen tube had formed (*Figure 7D and D'*), in *UAS-Fog x fkhGal4* embryos the apically constricting cells had formed a large shallow depression across the whole dorsal part of the placode, with the beginning of a deeper invagination at the dorsal-posterior corner (*Figure 7F and F'*). Following on from this, at stage 12, in contrast to the narrow opening to the invagination in wild-type placodes (*Figure 7E*), in *UAS-Fog x fkhGal4* placodes this whole section usually formed a deepened very wide pit (*Figure 7G*). This widened tube in Fog-overexpressing embryos, at stages 15–16 when all cells had invaginated, displayed a significantly enlarged and flattened lumen compared to wild-type tubes (*Figure 7H–J*). Interestingly, overexpression of Fkh using *UAS-Fkh x fkhGal4* led to a similarly enlarged area of apical constriction early on and invaginated tubes with aberrant and inflated lumens at late stages (*Figure 7—figure supplement 2*).

We set out to explain the link between an enlarged or widened invagination pit in the placode early during tube formation and the expanded lumen tube observed at late stages by considering why the invagination pit forms in its eccentric position in wild-type placodes. We formulated a simplistic theoretical consideration that assumes that following the specification of an initial group of cells that apically constrict and form the initial invagination (*Figure 7K–M*), neighboring coronae of cells will follow and invaginate (*Figure 7K'–M'*). Using an idealized version of apices of a wild-type placode with an eccentric pit, the number of cells invaginating, going from corona to corona, remains very steady and low (*Figure 7K'* and green curve in *Figure 7N*). In contrast, a centrally located pit (*Figure 7L*) would lead to an ever-increasing number of cells per corona that would need to invaginate (*Figure 7L'* and magenta curve in *Figure 7N*). An enlarged pit, a combination of the two previous scenarios and resembling aspects of the enlarged pit and invagination seen in *UAS-Fog x fkhGal4* embryos (*Figure 7M and M'*), would lead to the invagination of a large number of cells initially, followed by similarly large numbers in each corona (*Figure 7M'* and gray curve in *Figure 7N*). In fact, when counting invaginating coronae of cells in frames of time-lapse movies of wild-type (*Figure 7O*, green curve), *hkb⁻/⁻* (*Figure 7O*, magenta curve) curve and *UAS-Fog x fkhGal4* (*Figure 7O*, gray curve) embryos, the actual numbers observed were in very good agreement with the theoretical considerations outlined above (*Figure 7K–N*).

In the *hkb⁻/⁻* mutant embryos, the delayed central area of constriction turned into a widened invagination and the enlarged pit at late stages often collapsed into an enlarged crescent of subduction at the boundary (*Figure 3—figure supplement 2* and *Figure 3—video 3*; *Figure 3—video 4*), leading to malformed and enlarged lumen glands at the end of embryogenesis, as has been observed previously (*Myat and Andrew, 2002*).

Thus, the eccentrically placed invagination pit in the dorsal-posterior corner of the salivary gland placode is a key factor in ensuring the formation of a symmetrical narrow lumen tube during invagination, in part likely due to the control over the number of invaginating cells in each corona destined to internalize.

## Discussion

In this work, we investigated how the sustained morphogenesis of a tissue is achieved, in this case, the continued invagination of epithelial cells forming a tubular organ. We previously identified cell

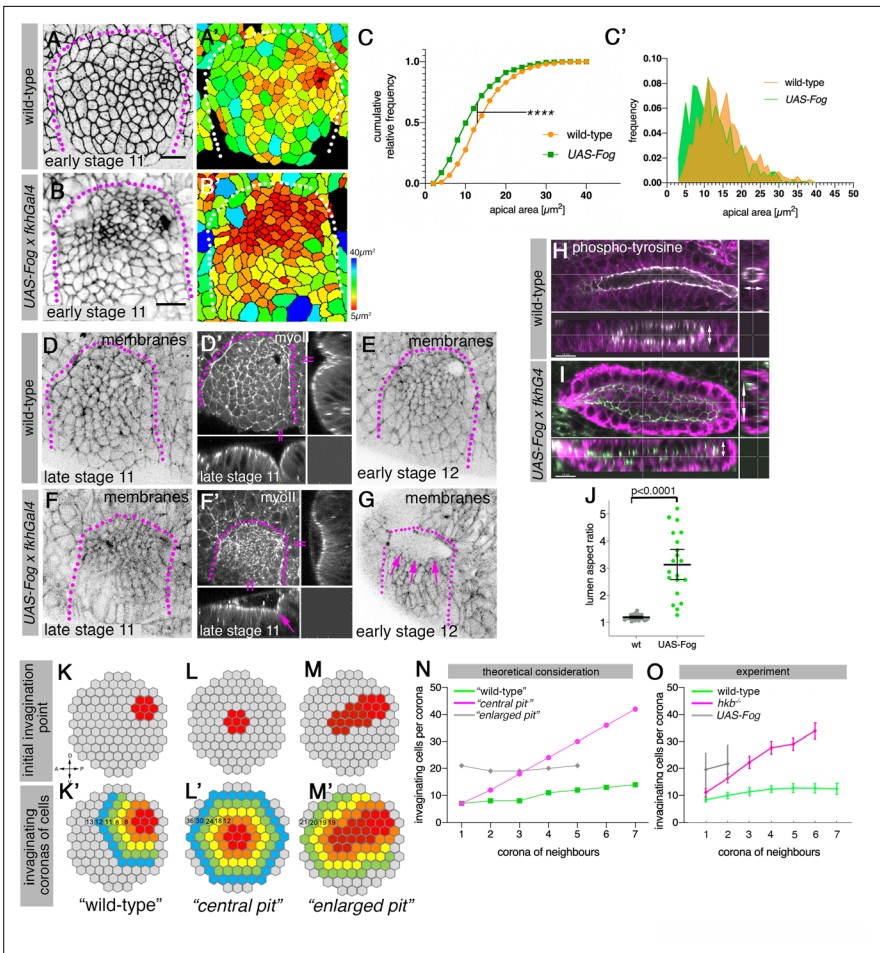

**Figure 7.** Changes to asymmetric placode patterning and invagination pit shape lead to misshapen organs. (**A–C'**) Whereas in wild-type placodes (**A, A'**) apical constriction commences only at the dorsal-posterior corner prior to the start of pit invagination, in placodes where Fog is overexpressed (using *UAS-Fog x fkhGal4*; **B, B'**) excess apical constriction occurs all across the dorsal side of the placode. (**A, B**) show cell apices labeled by anti-phospho-tyrosine labelling, (**A', B'**) show apical area size. Colored scale for cell area as in *Figure 1*. (**C, C'**) Widening the Fog expression domain leads to significantly more constricted apices: cell area distribution of *UAS-Fog x fkhGal4* differs from wild-type (Kolmogorov-Smirnov test, p<0.0001). Number of cells: wild-type, n=771 from seven embryos; *UAS-Fog x fkhGal4*, n=788 from five embryos. (**D–G**) In contrast to the narrow symmetrical pit and early invagination in wild-type placodes (**D-D'**), in *UAS-Fog x fkhGal4* placodes cells in the whole Fog-overexpression domain constrict and initially start to form a large shallow depression (**F, F'**). (**D–G**) are stills from time-lapse movies, with (**D'**) and (**F'**) showing surface view and xz/yz-cross sections, the arrow in (**F'**) points to the corner from where cells start to invaginate. In comparison to the small opening of the invagination observed in wild-type placodes at stage 12 (**E**), the initial large depression is still present in *UAS-Fog x fkhGal4* placodes at stage 12 whilst cells invaginate through a large pit ((**G**); magenta arrows). (**H–J**) At late stage 15, early stage 16, when salivary gland invagination and morphogenesis has finished, the final shape, and in particular lumen shape, of salivary gland cells overexpressing Fog (**I**); (*UAS-Fog x fkhGal4*) is altered compared to wild-type controls (**H**). Apical adherens junctions are marked by phospho-tyrosine labelling (white). Glands are shown in three orthogonal cross-sections. (**J**) Quantification of lumen aspect ratio, allowing identification of altered tube shapes such as widened tubes (Mann-Whitney test, p<0.0001; wild-type: n=23 embryos, *UAS-Fog x fkhGal4* n=20 embryos). (**K–O**) Theoretical considerations and experimental test on how altering the shape and size of the original invagination pit will affect the geometry and shape of the invaginating and invaginated tube. (**K, K'**) In the wild-type, a focussed eccentric pit (red in (**K**)) leads to similar numbers (**K'**) of cells invaginating within each corona even in the absence of any cell rearrangements (green curve in (**N**)). (**L, L'**) A central pit would lead to an increasing number of cells invaginating in each corona, leading to a widening tube (magenta curve in (**N**)). (**M, M'**) An enlarged initial pit would lead to an increased but steady number of cells (**M'**) invaginating (gray curve in (**N**)), again leading to a tube with enlarged lumen. Numbers of cells per corona are indicated. (**O**) Experimental test of the number of cells per invaginating corona in wild-type

*Figure 7 continued*

placodes or when the pit is central (*hkb*$^{-/-}$ mutant embryos) or the pit is enlarged, covering eccentric and central position (*UAS-Fog x fkhGal4* embryos). Shown are mean and SD of n=30, n=11, and n=5 for wild-type, *hkb*$^{-/-}$ mutant and *UAS-Fog x fkhGal4* embryos, respectively. See also *Figure 7—figure supplements 1 and 2*.

The online version of this article includes the following figure supplement(s) for figure 7:

**Figure supplement 1.** Fog overexpression in the salivary gland placode.

**Figure supplement 2.** Overerepression of Fkh in the salivary gland placode.

behaviors that become established across the placode and drive the initial tissue bending (*Sanchez-Corrales et al., 2018*).

A number of mechanisms appear at play that in combination ensure that a symmetrical and narrow-lumen tube is formed from an initially flat epithelial primordium. A temporal and spatial component is provided by the dynamically changing pattern of the transcription factors Hkb and Fkh. These are expressed initially with the highest protein level at the point where the invagination pit will form, before expression expands further across the placode over time. This ensures that the cells in the dorsal-posterior corner are the first to experience high levels of activity of these transcription factors, and we show that one of the important targets, the GPCR-ligand Fog, is found at high levels in these cells in a Hkb- and Fkh-dependent manner with a slight temporal delay. A further spatial component is provided by the fact that, as cells continuously internalize to form the tube on the inside of the embryo, new coronas of cells are brought into a 'near the pit' position and adjust their behavior accordingly. This position-specific behavior of apical constriction is most likely triggered by the previous temporal expansion of Fkh and also Hkb expression to these cells, thereby priming them for their next step in the path to tube formation. Interestingly, though, the pattern of Fkh and Hkb expression from t=0 min onwards, though generally expanding across the placode, is not identical. This could suggest that downstream targets begin to diverge at later stages. The *hkb*$^{-/-}$ mutant analysis in particular suggests that the central cells in the placode that manage to constrict in the absence of Hkb might only require Fkh activity to initiate Fog expression and function. These cells therefore undergo their normal apical constriction once Fkh expression has reached the central position. As they now are the first cells to constrict and invaginate, the *hkb*$^{-/-}$ mutants show a central and delayed constriction and invagination.

Our work strongly suggests that the combined early patterning of Hkb and Fkh expression defines the early invagination pit and its eccentric position. How though is Hkb and Fkh expression initially limited to this point? Both are expressed in the placode downstream of the homeotic factor Sex Combs Reduced (Scr; *Myat and Andrew, 2000b*; *Panzer et al., 1992*). They are also not the only factors expressed near exclusively at this corner first. At the transcription factor level in particular one of the parasegmentally repeated stripes of Wingless (Wg) expression overlays the posterior part of the placode, its expression domain defined by previous parasegmental patterning (*Clark, 2017*). And at the time point when Hkb and Fkh expression at the future pit location commences, the Wg expression within the placode region is also confined to this corner only, rather than extending all the way to the ventral midline (data not shown). Future studies will show whether a combination of Scr and Wg transcription factor activity could activate early pit-defining expression of factors such as Fkh and Hkb. In addition to transcription factors, some potential downstream effectors also appear to be expressed near the future pit location first, these include Btk29/Tec29 (*Chandrasekaran and Beckendorf, 2005*), Crumbs (*Myat and Andrew, 2002*; *Röper, 2012*), 18-Wheeler (*Kolesnikov and Beckendorf, 2007*), and Klarsicht (*Myat and Andrew, 2002*). Cell-based genomics approaches such as single-cell RNAseq or ATAC-seq should in the future allow to precisely determine the genetic program that underlies the regionalization of the placode.

Activation of actomyosin contractility through GPCR signaling driven by the autocrine secretion of the ligand Fog is a module repeatedly used during tissue bending and invagination events in *Drosophila* embryogenesis. What differs are the upstream activating transcription factors: during mesoderm invagination Fog expression on the ventral side is controlled by Twist and Snail (*Morize et al., 1998*), whereas its expression during posterior midgut invagination is controlled by Tailless and Hkb (*Weigel et al., 1990*). The effect of Hkb on Fog expression is likely direct, as the Fog locus contains predicted Hkb-binding sites (using oPOSSUM, http://opossum.cisreg.ca/oPOSSUM3/). The identity of the GPCR that could mediate the Fog signal in the salivary gland placode is unclear thus far.

Published and database (Flybase) in situs for the GPCRs previously reported to be involved in mesoderm invagination and germband extension, Mist and Smog (*Kerridge et al., 2016*; *Manning et al., 2013*), do not show an enriched expression in the placode. But with around 200 *Drosophila* GPCRs in existence, half of them orphan receptors (*Hanlon and Andrew, 2015*), the placode might well express yet another one, and we will focus on candidates previously identified in screens (*Maybeck and Röper, 2009*) and genomics approaches (our unpublished data) in the future.

The question of how cell behavior is modulated over time after the initial transcriptional patterning that defines primordium identity is key to understand general mechanisms that allow continuous organ formation. Different solutions seem to have been adopted in different tissues. Recent work analyzing mesoderm invagination in *Drosophila* has shown that the key morphogenetic effectors T48 and Fog show a gradient of expression due to differential timing of transcriptional activation, leading to different spatial accumulations of the transcripts (*Lim et al., 2017*) and proteins (*Heer et al., 2017*). Flattening the gradient experimentally (by expansion of the ventral domain) leads to a widened and perturbed invagination process. By contrast, the invagination of the posterior midgut or endoderm primordium in the fly embryo does not require sustained transcription, but instead a wave of apical constriction that is initiated by Fog expression in a subset of cells, appears to be propagated via mechanical feedback (*Bailles et al., 2019*). The salivary gland placode seems to follow a model of a temporal expansion or wave of transcriptional activity translating into positionally fixed behaviors. Nonetheless, our quantitative analysis of 3D cell behaviors strongly indicated that mechanical feedback does also play a role (*Sanchez-Corrales et al., 2018*). Which combination of mechanisms is ultimately at work to sustain a morphogenetic process could depend on a number of factors including the developmental stage, the speed of the process (consider ~15 min for mesoderm invagination versus ~30 min for endoderm invagination versus ~1.5 hr for secretory cell invagination in the salivary gland placode), as well as the influence of other nearby concomitant morphogenetic events.

The eccentric pit setup of the placode we suggest is key to wild-type tube formation with a narrow lumen, and we show that our simple geometrical considerations are in good agreement with the actual in vivo data of number of cells per corona invaginating. Whether the eccentricity only allows to control cell numbers of invaginating coronae, or whether such setup could also have further beneficial effects on for instance mechanical aspects of the tissue will be something we will actively explore both in vivo and in silico in the future. Curiously and maybe tellingly, the asymmetry of the placodal setup is retained in another tube formation process, the invagination of the posterior spiracles (*Sidor et al., 2020*; *Simões et al., 2006*). The posterior spiracle placode, like the salivary gland placode, is a flat epithelium with an eccentric invagination point. In this case, an initial tube is already present due to the previous invagination of the most posterior tracheal placode. Whether the asymmetrical setup is thus a conserved feature of further tube budding processes during development will be an exciting question to address. 3D mathematical modelling of tube morphogenesis assuming an eccentric versus centrally positioned invagination point will help to dissect the topological advantages of being asymmetric.

In summary, we have quantitatively unraveled a dynamic spatio-temporal patterning of transcription factors and switches in cell behaviors leading to positionally fixed behaviors during the morphogenesis of simple tubular organ. We foresee that such mechanisms will be important for establishing general mechanisms and morphogenetic modules at work during the morphogenesis of more complex tubular organs and potentially also for engineered tissues in a dish.

## Materials and methods

**Key resources table**

| Reagent type (species) or resource | Designation | Source or reference | Identifiers | Additional information |
|---|---|---|---|---|
| Gene (*Drosophila melanogaster*) | Forkhead | GeneBank | FLYB:FBgn0000659 | |
| Gene (*D. melanogaster*) | Non-muscle myosin II / sqh | GeneBank | FLYB:FBgn0003514 | |

*Continued on next page*

*Continued*

| Reagent type (species) or resource | Designation | Source or reference | Identifiers | Additional information |
|---|---|---|---|---|
| Genetic reagent (*D. melanogaster*) | *Armadillo-YFP* | CPTI collection PMID:25294944 | | |
| Genetic reagent (*D. melanogaster*) | *fkh-Gal4* | PMID:10625560 | | |
| Genetic reagent (*D. melanogaster*) | *UAS-Fog* | PMID:16123312 | | Gift from Thomas Lecuit |
| Genetic reagent (*D. melanogaster*) | *UAS-Cre* | Bloomington Stock Center BDSC:55801 | | |
| Genetic reagent (*D. melanogaster*) | *UAS-Brainbow* | PMID:22446736 | | Gift from Stefan Luschnig |
| Genetic reagent (*D. melanogaster*) | *sqh[AX3]; sqh::sqhGFP42* | PMID:14657248 | | |
| Genetic reagent (*D. melanogaster*) | *sqh[AX3];::sqhGFP42, UbiRFP-CAAX* | Kyoto *Drosophila* Genomic Research Centre Number 109822 PMID:30015616 | | |
| Genetic reagent (*D. melanogaster*) | *Scribble-GFP* | Kyoto *Drosophila* Genomic Research Centre | | |
| Genetic reagent (*D. melanogaster*) | *Venus-Hkb* | This study | | See Materials and methods section for details |
| Genetic reagent (*D. melanogaster*) | *Fkh-GFP* | Bloomington Stock Center BDSC:43951 | | |
| Genetic reagent (*D. melanogaster*) | *fkh[6]* | Bloomington *Drosophila* Stock Center, PMID:2566386 | FLYB:FBal0004012 | |
| Genetic reagent (*D. melanogaster*) | *hkb[2]* | Bloomington *Drosophila* Stock Center BDSC: 5457 | | |
| Antibody | Anti-DE-Cadherin (rat monoclonal) | Developmental Studies Hybridoma Bank at the University of Iowa | DSHB:DCAD2 | (1:10) |
| Antibody | PY-20, *P11120 (mouse monoclonal) | Transduction Laboratories | | (1:500) |
| Antibody | Anti-Fog (rabbit polyclonal) | PMID:24026125 | | (1:1000) |
| Antibody | Anti-Forkhead (guinea pig polyclonal) | PMID:2566386 | | (1:2000) |
| Antibody | Anti-Huckebein (rat polyclonal) | PMID:9056782 | | (1:500) |
| Antibody | Alexa Fluor 488/549/649-coupled secondary antibodies | Molecular Probes | | (1:200) |
| Antibody | Cy3-, Cy5-coupled secondary antibodies | Jackson ImmunoResearch | | (1:200) |
| Software, algorithm | otracks | PMID:19412170, PMID:24914560 | | Software file (custom software written in IDL) |
| Software, algorithm | *nd-safir* | PMID:19900849 | | Denoising algorithm. Available at http://serpico.rennes.inria.fr/doku.php?id=software:nd-safir:index |

## Fly stocks and husbandry

The following transgenic fly lines were used: *Armadillo-YFP* (CPTI collection described in *Lye et al., 2014*), *sqhAX3; sqh::sqhGFP42* (*Royou et al., 2004*) and *fkhGal4* (*Henderson and Andrew, 2000*; *Zhou et al., 2001*) (kind gift of Debbie Andrew); *UAS-Fog* (kind gift of Thomas Lecuit); $y^1 w^* cv^1 sqh^{AX3}$;

**Table 1.** Embryo genotypes as presented in figures.

| Genotype | Figure | Chromosome | Experiment type |
|---|---|---|---|
| Armadillo-YFP | 1 | 1 | Live-imaging |
| Armadillo-YFP;; sqhGFP42, UbiRFP/+ | 2 | 1,3 | Live-imaging |
| sqh$^{AX3}$;sqhGFP; UbiRFP | 2 | 1,3 | Live-imaging |
| hkb[2] p', ScribGFP | 3 | 3 | Live-imaging |
| sqhGFP42, UbiRFP; hkb[2] | 3 | 2,3 | Live-imaging |
| UbiRFP; VenusHkb | 4 | 2,3 | Live-imaging |
| UbiRFP; VenusHkb | 4 | 2,3 | Fixed samples |
| FkhGFP; wgGal4, UASpalmYFP, UbiRFP | 4 | 2,3 | Live-imaging |
| FkhGFP | 4 | 2 | Fixed samples |
| sqh$^{AX3}$;sqhGFP; UbiRFP | 5 | 1,3 | Fixed samples |
| sqhGFP42, UbiRFP; hkb[2] | 5 | 2,3 | Fixed samples |
| sqhGFP42, UbiRFP; fkh[6] | 5 | 2,3 | Fixed samples |
| sqh$^{AX3}$;sqhGFP; UbiRFP | 6 | 1,3 | Fixed samples |
| UAS-Fog; fkhGal4, sqhGFP42, UbiRFP | 6 | 2,3 | Fixed samples |

P[w$^{+mC}$ = sqh-GFP.RLC]C-42 M[w$^{+mC}$ = Ubi-TagRFP-T-CAAX]ZH-22A (Kyoto DGRC number 109822, referred to as *sqh$^{AX3}$;sqhGFP; UbiRFP*); [sqhGFP42,UbiRFP; fkh[6]/TM3 Sb Twi-Gal4::UAS-GFP] (*fkh[6]* allele from Bloomington); [hkb$^2$, p$^2$, Scribble-GFP]; [sqhGFP42,UbiRFP; hkb$^2$, p$^2$/ TM3 Sb Twi-Gal4::UAS-GFP] *hkb$^2$* allele from Bloomington BDSC: 5457, [*UbiRFP, Venus-HKB*] (this study), *FkhGFP* (Bloomington BDSC:43951) and [*FkhGFP; wgGal4, UAS-palmYFP*] (generated from membrane Brainbow; *Förster and Luschnig, 2012*; *Hampel et al., 2011*). See *Table 1* for details of genotypes used for individual figure panels.

- The CRISPR *Venus-HKB* was created using the following gRNA targets flanking the N-terminus of Hkb:
- CACCGCAAACCTACTCGCGACTT.
- A linker sequence between the inserted fluorescent protein and the protein was also used:
- GGAGGCGGAGGCTCGGGAGGCGGAGGCTCG.

## Embryo immunofluorescence labeling, confocal, and time-lapse

Embryos were collected on apple juice-agar plates and processed for immunofluorescence using standard procedures. Briefly, embryos were dechorionated in 50% bleach, fixed in 4% formaldehyde, and stained with primary and secondary antibodies in PBT (PBS plus 0.5% bovine serum albumin and 0.3% Triton X-100). Anti-Hkb was a gift from Chris Doe (*McDonald and Doe, 1997*) anti-Fkh was a gift from Herbert Jäckle (*Weigel et al., 1989*) and anti-Fog was a gift from Naoyuki Fuse (*Fuse et al., 2013*), PY-20, (*P11120, Transduction Laboratories); anti-CrebA (DSHB). Secondary antibodies used were Alexa Fluor 488/Fluor 549/Fluor 649 coupled (Molecular Probes) and Cy3 coupled (Jackson ImmunoResearch Laboratories). Samples were embedded in Vectashield (Vector Labs).

Images of fixed samples were acquired on an Olympus FluoView 1200 or a Zeiss 780 Confocal Laser scanning system as z-stacks to cover the whole apical surface of cells in the placode. Z-stack projections were assembled in ImageJ or Imaris (Bitplane), 3D rendering was performed in Imaris.

For live time-lapse experiments (see *Table 1*), embryos were dechorionated in 50% bleach and extensively rinsed in water. Stage 10 embryos were manually aligned and attached to heptane-glue coated coverslips and mounted on custom-made metal slides; embryos were covered using halocarbon oil 27 (Sigma-Aldrich) and viability after imaging after 24 hr was controlled prior to further data analysis. Time-lapse sequences were imaged under a 40×/1.3NA oil objective on an inverted Zeiss 780 Laser scanning system, acquiring z-stacks every 0.58–3 min with a typical voxel xyz size of 0.22×0.22×1 µm$^3$. Z-stack projections to generate movies in Supplementary Material were assembled

in ImageJ or Imaris. The absence of fluorescent *Twi-Gal4::UAS-GFP* was used to identify homozygous *hkb[2]* and *fkh[6]* mutant embryos. The membrane channel images from time-lapse experiments were denoised using *nd-safir* software (*Boulanger et al., 2010*).

## Cell segmentation and tracking

Cell segmentation and tracking were performed using custom software written in *Blanchard et al., 2009*. First, the curved surface of the embryonic epithelium was located by draping a 'blanket' over all image volumes over time, where the pixel-detailed blanket was caught by, and remained on top of binarised cortical fluorescence signal. Quasi-2D image layers were then extracted from image volumes at specified depths from the surface blanket. This step permitted to account for the curvature of the embryos in these quasi-2D projections. We took maximum intensity projections of a small number of near-surface image layers typically at 0–4 µm from membrane channels, for cell-apical analyses, apico-medial myosin and anti-Fog fluorescence channels and projections of image layers typically from 5 to 16 µm encompassing nuclear fluorescence for Venus-HKB and Fkh-GFP analyses. Membrane channels were filtered with median, top-hat, or high/low frequency filters as necessary to optimize subsequent cell tracking.

Cells in membrane channels were segmented using an adaptive watershedding algorithm as they were simultaneously linked in time. Manual correction of segmented cell outlines was performed for all fixed and time-lapse data. The segmentation of all the movies used in this study was manually corrected to ensure at least  90% tracking coverage of the placode at all times. Tracked cells were subjected to various quality filters (lineage length, area, aspect ratio, and relative velocity) so that incorrectly tracked cells were eliminated prior to further analysis. The number of embryos analyzed and the number of cells can be found in *Figure 1—figure supplement 1*, *Figure 3—figure supplement 1*.

## Mobile radial coordinate system for the salivary gland placode

Wild-type movies were aligned in time using as t=0 min, the frame just before the first sign of invagination at the future tube pit was evident. $hkb^{-/-}$ mutants were aligned using as a reference of embryo development the level of invagination of the tracheal pits that are not affected in the $hkb^{-/-}$ mutant as well as other morphological markers such as appearance and depth of segmental grooves in the embryo. Cells belonging to the salivary placode (without the future duct cells that comprise the two most ventral rows of cells in the primordium) were then manually outlined at t=0 min using the surrounding myosin II cable as a guide and ramified forwards and backwards in time. Only cells of the salivary gland placode were included in subsequent analyses.

At t = 0 min, the center of the future tube pit was specified manually as the origin of a radial coordinate system, with radial distance (in µm) increasing away from the pit. Circumferential angle was set to zero toward posterior, proceeding anti-clockwise for the placode on the left-hand side of the embryo, and clockwise for the placode on the right so that data collected from different sides could be overlaid.

The radial coordinate system was 'mobile,' in the sense that its origin tracked the center of the pit, forwards and backwards in time, as the placode translated within the field of view due to embryo movement or to on-going morphogenesis.

As calculated previously (*Sanchez-Corrales et al., 2018*), we employ a radial coordinate system centered at the invagination point at t=0 min. We specify the 'near to the pit' region as the region located between the invagination point and up to  33% of the distance between the invagination point and the far edge of the placode (with this region usually being within 13.7±3 µm of the pit), and with the rest of the cells defined as 'far from the pit.'

## Morphogenetic strain rate analysis

Detailed spatial patterns of the rates of deformation across the placode and over time quantify the net outcome of active stresses, viscoelastic material properties, and frictions both from within and outside the placode. We quantified strain (deformation) rates over small spatio-temporal domains composed of a focal cell and one corona of immediate neighbors over a ~ 3-min interval (*Blanchard et al., 2009*) and reviewed in *Blanchard, 2017*. On such 2D domains, strain rates are captured elliptically, as the

strain rate in the orientation of greatest absolute strain rate, with a second strain rate perpendicular to this.

For the early morphogenesis of the salivary gland placode, in which there is no cell division or gain/loss of cells from the epithelium, three types of strain rates can be calculated. First, total tissue strain rates are calculated for all local domains using the relative movements of cell centroids, extracted from automated cell tracking. This captures the net effect of cell shape changes and cell rearrangements within the tissue, but these can also be separated out. Second, domain cell shape strain rates are calculated by approximating each cell with its best-fit ellipse and then finding the best mapping of a cell's elliptical shape to its shape in the subsequent time point, and averaging over the cells of the domain. Third, intercalation strain rates that capture the continuous process of cells in a domain sliding past each other in a particular orientation, are calculated as the difference between the total tissue strain rates and the cell shape strain rates of cells. Strain rates were calculated using custom software written in L3Harris Geospatial IDL (code provided in *Blanchard et al., 2009* or by email from G.B.B.).

The three types of elliptical strain rates were projected onto our radial coordinate system (*Sanchez-Corrales et al., 2018*) using the eccentric pit location in wild-type (dorsal-posterior corner) or the center of the misplaced invagination point in $hkb^{-/-}$ mutants. The radial coordinate system permitted us to analyze radial and circumferential contributions. Strain rates in units of proportional size change per minute can easily be averaged across space or accumulated over time. We present instantaneous strain rates over time for spatial subsets of cells in the placode (*Figure 4—figure supplement 1*), and cumulative strain ratios for the same regions over time. These plots were made from exported data as done previously in *Sanchez-Corrales et al., 2018*. To test for differences in instantaneous strain rates of wt and $hkb^{-/-}$ time-lapse movies we used a mixed-effects model implemented in R (lmer4 package as in *Butler et al., 2009*; *Lye et al., 2015*; *Sanchez-Corrales et al., 2018*) with a significance threshold of $p < 0.05$. The phenotype (wt or $hkb^{-/-}$) was considered a fixed effect while the variation between embryos from the same phenotype was considered a random effect.

Cell shape elongation (*Figure 4H*) was calculated by fitting an ellipse into each cell. The two axes of each ellipse were projected into radial coordinates and the ratio of them was calculated.

## Neighbor exchange analysis

We used changes in neighbor connectivity in our tracked cell data to identify neighbor exchange events (T1 processes). Neighbor exchange events were defined by the identity of the pair of cells that lost connectivity in $t$ and the pair that gained connectivity at $t+1$. The orientation of gain we defined as the orientation of the centroid-centroid line of the gaining pair at $t+1$. We further classified gains as either radially or circumferentially oriented, depending on which the gain axis was most closely aligned to locally. We did not distinguish between solitary T1s and T1s involved in rosette-like structures.

From visual inspection, we knew that some T1s were subsequently reversed, so we characterized not only the total number of gains in each orientation but also the net gain in the circumferential axis, by subtracting the number of radial gains. Furthermore, when comparing embryos and genotypes, we controlled for differences in numbers of tracked cells by expressing the net circumferential gain per time step as a proportion of half of the total number of tracked cell-cell interfaces in that time step. We accumulated numbers of gains, net gains, and proportional rate of gain over time for WT and $hkb$ embryos (*Figure 4D and E*). Two sample Kolmogorov-Smirnov tests were used to determine significance at $p < 0.05$ for data in *Figure 3*.

## Automated medial myosin II quantification and oscillatory behavior and polarity

We extracted a quasi-2D layer image from the myosin II channel at a depth typically from 0 to 3 μm that maximized the capture of apical-medial myosin. We background-subtracted the myosin images and quantified the average intensity of apical-medial myosin as the fluorescence inside the segmented cell, excluding the fluorescence along cell-cell interfaces encompassing the cell edge pixels plus two pixels in a perpendicular direction on either side (considered junctional myosin).

The analysis of apical-medial fluorescence fluctuations was performed as described before in *Blanchard et al., 2010* and *Booth et al., 2014*. Briefly, the apical-medial fluorescence values in the

time series were detrended by subtracting a boxcar average with a window of 6 min (larger than the maximum expected fluctuation cycle length). All peaks and troughs were then identified in the fluorescence signal. Peaks or troughs that were associated with very small amplitude cycles were skipped. The remaining peaks and troughs were used to calculate the amplitude and the cycle length (peak to peak) and the amplitude (average peak to trough) of fluctuations.

The analysis of junctional myosin polarity was performed as described before in *Sanchez-Corrales et al., 2018*; *Tetley et al., 2016*. Briefly, the average junctional fluorescence intensity is considered a periodic signal around the cell and decomposed into individual components (modes) using Fourier analysis. The strength of myosin bipolarity corresponds to the amplitude of the second mode while the strength of unipolarity corresponds to the amplitude of the first mode. Similarly, the phase of modes 1 and 2 (uni-/bipolarity) represents the orientation of cell polarity. The uni- and bipolarity amplitudes are expressed as a proportion of the mean cell perimeter fluorescence. Cells at the border of the placode that are part of the supra-cellular actomyosin cable at the boundary were excluded from the analysis.

## Hkb, Fkh, and Fog fluorescence intensity quantifications

Average fluorescence was calculated after background subtraction as the mean fluorescence inside a segmented cell excluding the cell edge pixels and two pixels in a perpendicular direction on either side. Fixed samples of *Venus-Hkb* and *FkhGFP* embryos of mid to late stage 10 (t<<0 min), early stage 11 (t+0 min) and late stage 11 (t>>0 min) were pooled together with relevant time points from live-imaging as shown in *Table 1* and figure legends. Fluorescence was normalized in each image by dividing the average fluorescence of each cell by the 98th percentile value. Calculations were performed in R.

## Acknowledgements

The authors would like to thank the following people; for help with an experiment: Ghislain Gillard; for reagents and fly stocks: Debbie Andrew, Thomas Lecuit, Stefan Luschnig, Herbert Jäckle, and Naoyuki Fuse. The authors thank members of the lab for input on the manuscript. KR and YSC were supported by the Medical Research Council (file reference number U105178780). GB was supported by a Wellcome Trust Investigator Award (207553/Z/17/Z) to Bénédicte Sanson, Department of PDN, Cambridge, UK.

## Additional information

### Funding

| Funder | Grant reference number | Author |
|--------|------------------------|--------|
| Medical Research Council | U105178780 | Yara E Sánchez-Corrales Katja Röper |
| Wellcome Trust | 207553/Z/17/Z | Guy B Blanchard |

The funders had no role in study design, data collection and interpretation, or the decision to submit the work for publication.

### Author contributions

Yara E Sánchez-Corrales, Conceptualization, Formal analysis, Investigation, Methodology, Visualization, Writing – original draft; Guy B Blanchard, Conceptualization, Formal analysis, Investigation, Methodology, Software, Visualization, Writing – original draft; Katja Röper, Formal analysis, Methodology, Software, Writing – original draft

### Author ORCIDs

Yara E Sánchez-Corrales ⓘ http://orcid.org/0000-0003-1438-1994
Guy B Blanchard ⓘ http://orcid.org/0000-0002-3689-0522
Katja Röper ⓘ http://orcid.org/0000-0002-3361-766X

Decision letter and Author response
Decision letter https://doi.org/10.7554/eLife.72369.sa1
Author response https://doi.org/10.7554/eLife.72369.sa2

## Additional files

### Supplementary files
• Transparent reporting form

### Data availability
All data generated and analysed in the manuscript are included in the manuscript and supporting files.

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
