## [Editor Report]

This paper addresses a fundamental question in developmental biology, that is, how morphogenetic movements driving tissue folding are patterned to occur with the correct spatiotemporal dynamics. By correlating dynamic patterns of transcription factor expression with rigorous, quantitative analyses of cell behaviors across the salivary gland primordium in *Drosophila*, their results suggest Hkb and Fkh transcription factor patterning induces switches in cell behaviors at fixed positions to promote continued morphogenesis of the tubular structure. This mechanism is likely to be more generally important for the development of complex tubular organs.

---

## [Decision Letter]

**Decision letter after peer review:**

Thank you for submitting your article "Correct regionalisation of a tissue primordium is essential for coordinated morphogenesis" for consideration by *eLife*. Your article has been reviewed by 3 peer reviewers, one of whom is a member of our Board of Reviewing Editors, and the evaluation has been overseen by Anna Akhmanova as the Senior Editor. The following individual involved in review of your submission has agreed to reveal their identity: Adam C Martin (Reviewer #3).

Essential revisions:

1) The conclusion that the pattern of Hkb expression regulates radial cell behaviors is based on analyzing a loss-of-function mutant in hkb, rather than altering its expression pattern. However, complete loss of a transcription factor is likely to affect many cellular processes, not just actomyosin. An experiment that alters the pattern of gene expression in the domain (other than loss of the whole factor) is needed to support the authors claims. If existing Gal4-UAS tools permit, the authors should attempt to alter the expression pattern of Hkb or Fkh and determine the effect on actomyosin and cell behavior. If existing tools do not allow for a meaningful change in the expression pattern, the authors should alter their conclusions.

2) The conclusion that "changes to invagination pit size or position... perturb final organ shape.", is based on Fog overexpression driven by Fkh-Gal4. Since Fkh is what the authors propose to drive Fog expression in the proper pattern, it is important to show how endogenous Fog expression compares to Fog driven by Fkh-Gal4. Moreover, Fog overexpression is expected to overactivate contractility at center of the invagination in addition to changing its pattern. The authors should attempt to distinguish between effects of excessive contractility in central cells and expansion of the domain size.

3) To unambiguously demonstrate that cells far from the pit move toward the pit and then change behavior to apical constriction, a cell tracking analysis should accompany Figure 1, so that the reader can clearly identify that the same cells have moved and changed shape over time.

4) In general, the manuscript is quite densely written and one can easily get lost in the descriptions. The authors should simplify the description of their results, emphasizing the most important take home points. A short and simple conclusion should be drawn at the end of each Results section.

*Reviewer #1 (Recommendations for the authors):*

1. To unambiguously demonstrate that cells far from the pit move toward the pit and then change behavior to apical constriction, a cell tracking analysis should accompany Figure 1, so that the reader can clearly identify that the same cells have moved and changed shape over time.

2. A more detailed discussion of the timing of events should accompany the authors overall model that dynamic patterns of Fkh and Hkb expression regulate the pattern of Fog/myosin dynamics. Specifically, it appears that Fog expression, myosin localization, and apical constriction correlate both in space and time across the placode, whereas Fkh and Hkb expression patterns are offset by ~30-60 min. Considering the time it takes for transcription, translation, and secretion of Fog, the authors should discuss whether the timing of events are consistent with their model.

Other points

1. How are 'near the pit' and 'far from the pit' cells selected and defined? This should be explained in the main text or legend.

2. Figure 1C. y-axis. Please define pp/min.

3. Figure 2E. I'm a little confused how the medial myosin cycle is defined. Looking at the data from figure 2D, it looks like medial myosin peaks only once (green boxes) over the time course. Or is this because data in Figure 2D are averaged across all cells in the radial stripe, so the cycles are obscured? Also, if cells far from the pit lack medial myosin, what is being quantified for cells far from the pit in 2E and F (and also 2G and H?)?

4. Figure 4 was overall very hard to follow. I suggest moving A' and B' panels to the left so it is clear that A panels refer to cells near the pit and B panels away from the pit. Also, I think circumferential vs radial behaviors need to be clearly defined in the text, labeled on the figure (I appreciate the schematics, but I wasn't sure which was radial and which was circumferential until panel C), and discussed separately in the text. For panels K and L, a comparison to WT intercalation is needed to appreciate the defect in Hkb-/-.

5. Figure 4F and G. Please describe what is the y-axis for myosin polarity (pp/min? what is being measured here?).

6. Figure 6. Several image panels (but not all) show paired gland placodes in more zoomed out images compared to previous figures. To be consistent with prior figures, I suggest cropping the images to show individual placodes in panels D-I and M-O.

7. Figure 7. Fkh-Gal4 is used to overexpress Fog in an aberrant pattern. But since Fkh is what the authors propose to drive Fog expression in the proper pattern, it is important to show how endogenous Fog expression compares to Fog driven by Fkh-Gal4.

8. Figure 7. Please show paired panels for WT and UAS-Fog x Fkh-Gal4 with the same markers for D-H. For example, Crumbs is shown for the WT but membranes are shown for Fog overexpression. Also stages 12 and 13 are shown for Fog overexpression but not for WT.

9. Figure 7P compares experimental observations to the in silico model. Why are there only two coronae measured for the UAS-Fog condition, and 6-7 coronae for WT and Hkb-/-?

*Reviewer #2 (Recommendations for the authors):*

1. L 208. "hkb-/- mutants show a delayed symmetrical apical constriction". This part in particular should be reformulated to make it easier to digest. Maybe separating the apical constriction part from the intercalation part could make it easier. The title is not taking into account the whole analysis of cell intercalation.

2. L257-293. A short summary of the main observations described in this paragraph is missing at the end. This description for example "in the hkb-/- mutants cells specified at t = 0 min to be near the coordinates of the later position of central apical constriction" (L 272) is quite complicated and it took me a while to understand. The analysis of cell shape, tissue strain, and individual cell intercalation behavior is a little complex to follow.

3. L440. "Hence the expansion of Fkh protein to the center of the placode in the hkb-/- mutants might be the cause of the delayed constriction of cells here, whereas the earlier initial constriction at the site of the eccentric pit was absent in the mutant because it requires both Fkh and Hkb." I do not follow the idea here. I cannot see any change in the expression pattern of Fkh at any stage (as mentioned in the text), so how this can unchanged pattern could explain the delay observed in Hbk mutant?

*Reviewer #3 (Recommendations for the authors):*

1. The authors state in the abstract that "early transcriptional patterning of key morphogenetic transcription factors drives the selective activation of downstream morphogenetic modules, such as... that activates actomyosin activity". To support their claim the authors analyze transcription factor mutants, such as hkb-/-, which disrupts the pattern. However, complete loss of a transcription factor is likely to affect many cellular processes (i.e. not just actomyosin). There is no experiment that alters the pattern of gene expression in the domain (other than loss of the whole factor). The authors should discuss the limitations of this experiment a bit more or if they have a way to alter the expression pattern, include that data.

2. The conclusion that "changes to invagination pit size or position... perturb final organ shape.": To test this, the authors overexpress Fog, which is expected to overactivate contractility at center of the invagination in addition to changing its pattern. The authors did not distinguish between or characterize possible effects of excessive contractility in central cells and expansion of the domain size. For example, how much is contractility being elevated in pit cells.

3. Figure 6: Fog concentration clearly is anti-correlated with apical constriction, but it is not clear if this is cause or effect. For an apically secreted protein it is possible that apical constriction concentrates apical vesicles that contain this protein, leading to greater intensity. Authors should determine if the accumulation pattern precedes apical constriction.

4. The authors use the terms "near pit" and "far from pit" consistently throughout the paper. However, the authors do not clearly define how these positions were determined. For example, in Figure 5 the line seems to be at 13.7uM, but Figure 6 has it at slightly less than 13.7uM. The line should be defined in the text and made consistent in the figures.

---

## [Author Response]

Essential revisions:1) The conclusion that the pattern of Hkb expression regulates radial cell behaviors is based on analyzing a loss-of-function mutant in hkb, rather than altering its expression pattern. However, complete loss of a transcription factor is likely to affect many cellular processes, not just actomyosin. An experiment that alters the pattern of gene expression in the domain (other than loss of the whole factor) is needed to support the authors claims. If existing Gal4-UAS tools permit, the authors should attempt to alter the expression pattern of Hkb or Fkh and determine the effect on actomyosin and cell behavior. If existing tools do not allow for a meaningful change in the expression pattern, the authors should alter their conclusions.

We agree with the editor’s suggestion that ideally in addition to loss of function an alteration of spatial or temporal expression of Hkb and/or Fkh would be ideal. As we show, both Hkb and Fkh expressed first at the position where the future invagination point will form, and both are required for the early constriction at this position. In fkh^-/-^ no invagination occurs (Sanchez-Corrales et al., 2018), and in hkb^-/-^ a central invagination forms as shown here. We do not claim at all that it is Huckebein alone that drives the radial patterning of behaviours, only that is plays an important part in the patterning, and its loss leads to an interesting phenotype that helped us dissect how behaviours are governed.

So although we agree that overexpression of Hkb could lead to interesting effects, this is by no means guaranteed as it is not the only factor affecting patterning and invagination.

Furthermore, such overexpression experiments are unfortunately limited by the available tools. Firstly, using the UAS/Gal4 ectopic expression system, the only drivers spatially intersecting the placode in parasegment 2 of the embryo are wg-Gal4, leading to a stripe/patch of expression that unfortunately mostly covers the region where the invagination pit will form, and is thus not of use for ectopic Hkb/Fkh expression as these are already highly expressed in this region from early on. The other driver is en-Gal4 that leads to expression in a stripe at the anterior margin of the placode. In our experience, this driver in the placode only induces higher expression levels once the invagination has started, so the domain of endogenous Fkh and Hkb expression and function will already have expanded. This leaves overexpression in the whole placode using fkhGal4.

With regards to Hkb overexpression, we have attempted this, but the available UAS-Hkb transgenic fly line we obtained does not seem to actually drive increased Hkb expression, so the genetic background might have changed or the transgene lost. We received this stock from different sources, but with no luck.

By contrast, overexpressoin of Fkh using fkhGal4 x UAS-Fkh leads to highly aberrantly-shaped glands at late stages with an enlarged lumen, and too many apically-constricted cells earlier on, this is now shown in Supplemental Figure 6_2. As we detail in the manuscript, the central placodal cells only require Fkh to drive apical constriction (and therefore in the hkb^-/-^ these cells will constrict once Fkh is expressed here), and so early homogeneous expression of Fkh leads to a larger area of apical constriction similar to what we observed when Fog is overexpressed, again confirming that the ordered internalisation of cells in defined coronae allows the formation of the narrow symmetrical tube of the final organ.

2) The conclusion that "changes to invagination pit size or position... perturb final organ shape.", is based on Fog overexpression driven by Fkh-Gal4. Since Fkh is what the authors propose to drive Fog expression in the proper pattern, it is important to show how endogenous Fog expression compares to Fog driven by Fkh-Gal4. Moreover, Fog overexpression is expected to overactivate contractility at center of the invagination in addition to changing its pattern. The authors should attempt to distinguish between effects of excessive contractility in central cells and expansion of the domain size.

We thank the reviewer for pointing this out and are happy to provide clarification.

The conclusion cited above is based on both the overexpression of Fog across the whole placode as well as the hkb^-/-^ mutant phenotype, so not only on an overexpression experiment. However, we agree that identifying whether Fog is actually ectopically or at higher level expressed in fkhGal4 x UAS-Fog embryos is important.

The overexpression using the fkhGal4 driver requires a further explanation: fkhGal4 is using a 1kb promoter fragment of the fkh locus to drive overexpression. This fragment has previously been shown to be responsible for salivary gland specific expression of Fkh (Zhou, Bagri and Beckendorf, 2001, Dev.Biol., doi:10.1006/dbio.2001.0367), but represents only 10% of the so far identified regulatory regions of the fkh locus. Thus, expression driven by fkhGal4, though mostly specific to the salivary gland placode, does not mimic endogenous Fkh expression in pattern or timing. To illustrate this, we include a Author response image 1 to show srcGFP expression driven by fkhGal4 in comparison to antibody labeling for endogenous Fkh protein (with the one caveat that GFP folding an maturation most likely underestimates timing of expression of other factors using fkhGal4). fkhGal4 driven GFP expression starts off at mid stage 10 in a few central cells (A-B’’) but very quickly most placodal cells show very high levels of expression (C-D’’’); note that this is already at late stage 10, prior to any commencement of tissue bending (as can be seen in cross sections in C’’’ and D’’’).

**Author response image 1. sa2fig1:** Comparison of endogenous Fkh protein and expression of transgenes via fkhGal4. A-B’’ At mid stage 10, endogenous Fkh protein, revealed using an antibody against Fkh (red), is already spreading across the placode from the initial expression at the forming pit position (asterisks in all panels mark the position of the future pit). At this stage, a srcGFP transgene (green) expressed under fkhGal4 control can be clearly identified to begin to be expressed in central cells of the placode at varying levels. C-D’’’ At late stage 10, when endogenous Fkh protein (red) is seen in all secretory placodal cells but constriction near the forming invagination point is only just beginning and there is no tissue bending yet (see cross section panels in C’’’ and D’’’), srcGFP expression (green) driven by fkhGal4 is very strong in nearly all placodal cells, with only slightly lower levels in the most anterior cells. Note that the laser power used to reveal GFP was about 5 times higher for panels A-B’’ compared to panels C-D’’’. Cell outlines are marked by an antibody against junctional phosphotyrosine (PY20; blue).

In embryos that overexpress Fog (UAS-Fog) driven by fkhGal4, we have now stained with anti-Fog antibody and include this information as Supplemental Figure 6_1. In wild-type embryos we have shown and quantified in the manuscript (Figure 6) that Fog expression begins near the future pit site and then expands across the placode, correlating positively with average medial myosin II intensity and negatively with apical cell area. In contrast, when Fog is overexpressed already at late stage 10 (panels B-C’’ Supplemental Figure 6_1) Fog protein levels across the placode are very high and fairly homogeneous (similar to srcGFP in panels C-D’’’). Surface views and especially cross sections show that Fog antibody labeling is punctate throughout the more apical part of the cytoplasm, most likely corresponding to Fog-containing vesicles about to be secreted apically. Similar vesicles are also observed in the wild-type, but here only strongly at early stage 11 close to the forming invagination pit (A-A’’ Supplemental Figure 6_1).

Therefore, we conclude that the increased domain size of Fog expression most likely drives apical constriction in a larger number of cells and at an earlier timepoint than in the wild-type.

With respect to an increase in myosin activity itself, because many cells start bending the tissue and invaginate at the same time when Fog is overexpressed (in UAS-Fog x fkhGal4), segmentation of movies beyond the initial timepoints is near impossible. Hence the best proxy for changes in contractility is the intensity of apical-medial myosin. We have now quantified medial myosin intensity in embryos overexpressing UAS-Fog under fkhGal4 control in comparison to control embryos. We express this as the ratio of medial myosin to junctional myosin intensity to account for the fact that control embryos carry two copies of the sqhGFP transgene whereas fkhGal4 x UAS-Fog embryos only carry one copy (due to the combination of chromosomes required for the cross). This analysis showed that fkhGal4 x UAS-Fog cells in the placode show an 50% increase in medial myosin intensity. This is now included in Supplemental Figure 6_1 (as panel E) and discussed in the results. This increase could suggest stronger contractility, but as also many more cells constrict and internalise it is impossible to disentangle whether the aberrant process of invagination is mostly due to the change in cell number constricting or in addition also any change in strength of the constrictions. In terms of our theoretical considerations, overexpression of Fog clearly leads to an enlarged pit invaginating with a resultant aberrant and inflated lumen shape of the tubular organ. A detailed time-resolved and quantitative analysis of the Fog-overexpression phenotype would be interesting but due to the complex curvature of the forming enlarged pit is currently not possible with our existing methods.

3) To unambiguously demonstrate that cells far from the pit move toward the pit and then change behavior to apical constriction, a cell tracking analysis should accompany Figure 1, so that the reader can clearly identify that the same cells have moved and changed shape over time.

The tracking of all segmented placodal cells is inherent in our quantitative analysis, but was not explicitly shown, though in the original version of the manuscript it was shown implicitly in plots such as the now panel Figure 1E, Figure 4A-B’’. As we discuss below in response to Reviewer 1, point 1, cells are defined as near the pit and far from the pit based on their placodal position at t = 0 min and defined from our previous studies (Sanchez-Corrales et al., 2018). Therefore, cells ‘far from the pit’ beyond t = +20min are now all in very close proximity to the pit and are contracting. This is also shown in the strain rate analysis in Figure 4, where in panels B and B’ the solid green line is showing cumulative cell shape change and is tracking at a value of just above zero (for radial change, B) or below zero (for circumferential change, B’) until t = +20 min, when the rate of constriction increases drastically for the originally ‘far from the pit’ cells.

But we agree that it is important to show this behaviour and change in a more intuitive way and have therefore changed Figure 1 accordingly:

Panels B-D now show all placodal cells and only the cells defined as near and far from the pit (defined at t = 0 min) until all cells are internalised. B shows these identities and highlights four individual examples. C and D show the apical area of near and far from the pit cells, respectively, clearly visualising that only cells in a position of proximity to the pit are actively constricting.

Panel E, as before, shows the cumulative apical area change of cells defined as near to or far from the pit (defined at t = 0 min), illustrating in the grey highlighted part of the solid curve that cells that were far from the pit do not change apical area until about t = +20min when also these cells, that are now in close proximity to the pit, constrict as shown in D.

New panels in F show the evolution of the apical area size for the four example cells highlighted in B. This clearly shows that the two cells originally ‘near the pit’ rapidly constrict their apical area whereas the two cells originally ‘far from the pit’ do not change their area initially but then switch to a fast contraction mode after t = +18’ 21’’. For the blue ‘far from pit’ cell in F, the panels in G show its involvement in a cell intercalation event, a rosette formation and resolution, that leads to the overall contraction of the area covered by this group of cells in the circumferential direction and expansion in the radial direction (as the strain rates in Figure 4 also show).

We hope that these additions make the cell behaviours and changes therein clearer.

4) In general, the manuscript is quite densely written and one can easily get lost in the descriptions. The authors should simplify the description of their results, emphasizing the most important take home points. A short and simple conclusion should be drawn at the end of each Results section.

We have extensively rewritten the manuscript with this in mind and hope the revised version is less dense and clearer, though we posit that quantitative analysis of complex processes requires accurate description, and we don’t want to lose accuracy in favour of simplicity.

Reviewer #1 (Recommendations for the authors):1. To unambiguously demonstrate that cells far from the pit move toward the pit and then change behavior to apical constriction, a cell tracking analysis should accompany Figure 1, so that the reader can clearly identify that the same cells have moved and changed shape over time.

Please see our detailed response above.

2. A more detailed discussion of the timing of events should accompany the authors overall model that dynamic patterns of Fkh and Hkb expression regulate the pattern of Fog/myosin dynamics. Specifically, it appears that Fog expression, myosin localization, and apical constriction correlate both in space and time across the placode, whereas Fkh and Hkb expression patterns are offset by ~30-60 min. Considering the time it takes for transcription, translation, and secretion of Fog, the authors should discuss whether the timing of events are consistent with their model.

We agree that these are important considerations. We have previously analysed the literature for educated estimates of the rates that will affect the delay between transcription factor (Hkb, Fkh) activity in the nucleus and Fog secretion/binding to the GPCR followed by the downstream signalling cascade leading to myosin phosphorylation. We describe our estimations here and have also added this to the manuscript (lines 414-418):

“Estimates from mammalian cells as well as direct measurements in *Drosophila* syncytial embryos put the rate of transcription in the range of 1-2min/kb (Shamir et al., 2016, Cell 164:1302; Garcia et al., 2013, Current Biology 23:2140-2145). Translation is estimated to take about 10aa/s (about 1 min/300aa protein), with protein folding occurring co-translationally and in the order of ms-min (Shamir et al., 2016, Cell 164:1302). Secretion/movement through the secretory pathway again on average takes in the order of minutes, though this of course depends on the cargo. Once bound to the receptor, the signalling cascade leading to myosin phosphorylation should occur in the region of seconds. Hence, a delay of 30-60 min appears very compatible with Fkh and Hkb activity leading to fog transcription, translation, folding, secretion, binding to the receptor followed by signalling to drive myosin phosphorylation.”

Other points1. How are 'near the pit' and 'far from the pit' cells selected and defined? This should be explained in the main text or legend.

We are sorry if this was not well enough explained. As discussed above, we have now added the below definition to the main text as well as the legend in the modified Figure 1:

“As calculated previously (Sanchez-Corrales et al., 2018), we employ a radial coordinate system centred at the invagination point at t = 0 min. We specify the ‘near to the pit’ region as the region located between the invagination point and up to 33% of the distance between the invagination point and the far edge of the placode (with this region usually being within 13.7 +/- 3 µm of the pit), and with the rest of the cells defined as ‘far from the pit’.”

2. Figure 1C. y-axis. Please define pp/min.

This proportion per minute (pp/min). We have now defined this in all relevant figure legends.

3. Figure 2E. I'm a little confused how the medial myosin cycle is defined. Looking at the data from figure 2D, it looks like medial myosin peaks only once (green boxes) over the time course. Or is this because data in Figure 2D are averaged across all cells in the radial stripe, so the cycles are obscured? Also, if cells far from the pit lack medial myosin, what is being quantified for cells far from the pit in 2E and F (and also 2G and H?)?

In Figure 2E the average myosin cycle length for a cell in a given position at a given time is plotted as all cells go through many cycles. And yes, in Figure 2D, the reviewer is right in that these are data from all cells in that stripe pooled, quantifying the average medial myosin intensity, hence revealing the overall stronger apical-medial myosin intensity in the position near the pit. Figure 2D does not take into account individual cycles (which are presented in Figure 2 E-G). This type of analysis has been used by us and others in the past (Blanchard et al., 2010, Development; Fischer et al. 2014 PLoS One; Booth et al., 2014, Dev Cell; Machado et al., 2015, BMC Biol).

With regards to medial myosin intensity and fluctuations in ‘near the pit’ and ‘far from the pit’ cells (as defined at t = 0min), this split used for part of the analysis into two groups is of course to some extent arbitrary as biologically there is no such sharp divide, but as our analysis here shows it is a gradation. The splitting into two groups with predominant feature and behaviours that differ allows a meaningful quantitative analysis especially for the strain rate calculations. Splitting the cells into more groups (of biologically probably more meaningful shared characteristic and behaviours) will reduce the number of cells that we are able to track and analyse to an extent that will make statistical analyses impossible. Hence, there are always some cells included in the ‘far from pit’ group even at early time points that will show some medial myosin intensity and fluctuations. But as panel 3D-G show these cells show reduced intensity, longer cycle length and hence lower medial myosin strength and do not yet drive apical constriction.

4. Figure 4 was overall very hard to follow. I suggest moving A' and B' panels to the left so it is clear that A panels refer to cells near the pit and B panels away from the pit. Also, I think circumferential vs radial behaviors need to be clearly defined in the text, labeled on the figure (I appreciate the schematics, but I wasn't sure which was radial and which was circumferential until panel C), and discussed separately in the text. For panels K and L, a comparison to WT intercalation is needed to appreciate the defect in Hkb-/-.

We are sorry that this Figure was confusing and in the revised version have tried to make it more intuitively accessible.

By way of explanation, though, the strain rate analyses in A-B’’ are different to the analyses of individual cell behaviours in panels C onwards. For the strain rate analysis, as detailed in the Methods section ‘Morphogenetic strain rate analysis’, small domains of cells are analysed to capture the net effects of tissue shape changes as well as cell shape changes and the calculated intercalation rates. These changes are by definition always expressed as a strain rate along the orientation of the greatest absolute strain, with a second strain rate perpendicular to this. Because of the circular tissue geometry and focussed invagination point of the placode, any strain rates are then projected onto the radial coordinate system we use. This is the basis for the two components to any strain rate analysed and depicted in Figure 4, each strain has a circumferential and a perpendicular radial contribution. For analysing morphogenesis in other tissues, such analysis has used two perpendicular components of strain oriented along embryonic axes.

We have made sure to cite our initial analysis (Sanchez-Corrales, 2018) that has a detailed visual explanation of the strain rate analyses in Figure 2 A-B.

The oriented behaviours schematised and quantified in C-E due to the circularity and inherent biology of the tissue follow the same orientations of circumferential and radial, but these quantifications are based on individual events and individual cell junctions and how they evolve over time, rather than small domains of cells.

With regards to changes in the figure and manuscript, as suggested we have moved the explanatory panels to the start of the figure (now panels A and B), and we have added the words ‘radial’ and ‘circumferential’ to the small diagrams within the graphs in A’, A’’, B’, B’’.

5. Figure 4F and G. Please describe what is the y-axis for myosin polarity (pp/min? what is being measured here?).

The underlying method to quantify uni-and bi-polarity of junctional myosin (based on Tetley et al., 2016 and Sanchez-Corrales et al., 2018) is explained in the Methods section.

Within the panels, the uni- and bipolarity amplitudes are expressed as a proportion of the mean cell perimeter fluorescence, and their rate of change as proportion (pp)/min is hence plotted here. We have added the following sentence to the Figure 4 legend for panels F, G:

“Plotted are the rates of change of the uni- and bipolarity amplitudes as a proportion (pp)/min of the mean cell perimeter fluorescence.”

6. Figure 6. Several image panels (but not all) show paired gland placodes in more zoomed out images compared to previous figures. To be consistent with prior figures, I suggest cropping the images to show individual placodes in panels D-I and M-O.

We have adjusted this and now just show a single placode as in all other panels.

7. Figure 7. Fkh-Gal4 is used to overexpress Fog in an aberrant pattern. But since Fkh is what the authors propose to drive Fog expression in the proper pattern, it is important to show how endogenous Fog expression compares to Fog driven by Fkh-Gal4.

Please see our detailed response to the editor’s comments above.

8. Figure 7. Please show paired panels for WT and UAS-Fog x Fkh-Gal4 with the same markers for D-H. For example, Crumbs is shown for the WT but membranes are shown for Fog overexpression. Also stages 12 and 13 are shown for Fog overexpression but not for WT.

We have updated and changed panels in Figure 7, we now show the same membrane marker in comparable samples for wt and UAS-Fog, and show matched images for the different stages shown.

9. Figure 7P compares experimental observations to the in silico model. Why are there only two coronae measured for the UAS-Fog condition, and 6-7 coronae for WT and Hkb-/-?

Because due to the large number of cells invaginating in the UAS-Fog x fkhGal4, nearly all secretory cells are internalised in these two coronae, any remaining cells are impossible to track and segment with our current methods due to the strong curvature.

Reviewer #2 (Recommendations for the authors):1. L 208. "hkb-/- mutants show a delayed symmetrical apical constriction". This part in particular should be reformulated to make it easier to digest. Maybe separating the apical constriction part from the intercalation part could make it easier. The title is not taking into account the whole analysis of cell intercalation.

We have rewritten this section extensively to make it easier to understand. We have also split the section in two with new subheading, so the strain rate analysis and description of Figure 4 is separated from the discussion of data in Figure 3.

2. L257-293. A short summary of the main observations described in this paragraph is missing at the end. This description for example "in the hkb-/- mutants cells specified at t = 0 min to be near the coordinates of the later position of central apical constriction" (L 272) is quite complicated and it took me a while to understand. The analysis of cell shape, tissue strain, and individual cell intercalation behavior is a little complex to follow.

Please see comment above that also applies to this section.

3. L440. "Hence the expansion of Fkh protein to the center of the placode in the hkb-/- mutants might be the cause of the delayed constriction of cells here, whereas the earlier initial constriction at the site of the eccentric pit was absent in the mutant because it requires both Fkh and Hkb." I do not follow the idea here. I cannot see any change in the expression pattern of Fkh at any stage (as mentioned in the text), so how this can unchanged pattern could explain the delay observed in Hbk mutant?

As stated above:

We conclude from our data that the initial constriction at the eccentric position where the pit forms depends on both Fkh and Hkb. In the hkb^-/-^ mutant the central cells in the placode manage to constrict in the absence of Hkb likely only require Fkh activity to initiate Fog expression and function (and hence constriction). These cells therefore undergo their normal apical constriction once Fkh expression has reached the central position (as Fkh expression is unaffected in the hkb^-/-^ mutant as we show). As they now are the first cells to constrict and invaginate, the hkb^-/-^ mutants show a central and delayed constriction and invagination.

Reviewer #3 (Recommendations for the authors):1. The authors state in the abstract that "early transcriptional patterning of key morphogenetic transcription factors drives the selective activation of downstream morphogenetic modules, such as... that activates actomyosin activity". To support their claim the authors analyze transcription factor mutants, such as hkb-/-, which disrupts the pattern. However, complete loss of a transcription factor is likely to affect many cellular processes (i.e. not just actomyosin). There is no experiment that alters the pattern of gene expression in the domain (other than loss of the whole factor). The authors should discuss the limitations of this experiment a bit more or if they have a way to alter the expression pattern, include that data.

Please see our detailed response to the editor’s question (1) above.

2. The conclusion that "changes to invagination pit size or position... perturb final organ shape.": To test this, the authors overexpress Fog, which is expected to overactivate contractility at center of the invagination in addition to changing its pattern. The authors did not distinguish between or characterize possible effects of excessive contractility in central cells and expansion of the domain size. For example, how much is contractility being elevated in pit cells.

Please see our detailed response to the editor’s question (2) above.

3. Figure 6: Fog concentration clearly is anti-correlated with apical constriction, but it is not clear if this is cause or effect. For an apically secreted protein it is possible that apical constriction concentrates apical vesicles that contain this protein, leading to greater intensity. Authors should determine if the accumulation pattern precedes apical constriction.

We assume the reviewer means that Fog concentration is anti-correlated with apical area (not apical constriction).

At earliest stages analysed by us, and an example is shown in Figure 6A-C, Fog intensity is already higher in a small group of cells at the position where the pit will form, but there is not yet any concentrated constriction of cells here yet. Thus, we would conclude that Fog increase precedes constriction.

The same can in fact be seen in the overexpression of Fog, using UAS-Fog x fkhGal4 (Supplemental Figure 6_1 B-B’’) where ectopic Fog protein levels are very high (compared to what the wild-type would show at this stage) but cells have not overconstricted yet.

4. The authors use the terms "near pit" and "far from pit" consistently throughout the paper. However, the authors do not clearly define how these positions were determined. For example, in Figure 5 the line seems to be at 13.7uM, but Figure 6 has it at slightly less than 13.7uM. The line should be defined in the text and made consistent in the figures.

Please see above: we have now added an explicit definition of the terms to the results and the methods section.

With regards to the discrepancy between Figure 5 and 6 (i.e. where the dotted line in the graphs is place), this seems to be a mistake in assembling the panel in Figure 6. We thank the reviewer for noticing this and pointing it out!